# FeatureBench: Benchmarking Agentic Coding for Complex Feature Development

**Qixing Zhou**[1,2]*, **Jiacheng Zhang**[1,2]*, **Haiyang Wang**[2]*, **Rui Hao**[1], **Jiahe Wang**[1], **Minghao Han**[1,2]
**Yuxue Yang**[1], **Shuzhe Wu**[2] , **Feiyang Pan**[2] , **Lue Fan**[1†], **Dandan Tu**[2] , **Zhaoxiang Zhang**[1†]

[1] Institute of Automation, Chinese Academy of Sciences    [2] Huawei Technologies Co., Ltd

haiyang.wang@huawei.com    lue.fan@ia.ac.cn

 Code: github.com/LiberCoders/FeatureBench     Dataset: FeatureBench
 Project Page: Beyond bug fixing and ship real features.

## Abstract

Agents powered by large language models (LLMs) are increasingly adopted in the software industry, contributing code as collaborators or even autonomous developers. As their presence grows, it becomes important to assess the current boundaries of their coding abilities. Existing agentic coding benchmarks, however, cover a limited task scope, *e.g.*, bug fixing within a single pull request (PR), and often rely on non-executable evaluations or lack an automated approach for continually updating the evaluation coverage. To address such issues, we propose FeatureBench, a benchmark designed to evaluate agentic coding performance in end-to-end, feature-oriented software development. FeatureBench incorporates an execution-based evaluation protocol and a scalable test-driven method that automatically derives tasks from code repositories with minimal human effort. By tracing from unit tests along a dependency graph, our approach can identify feature-level coding tasks spanning multiple commits and PRs scattered across the development timeline, while ensuring the proper functioning of other features after the separation. Using this framework, we curated 200 challenging evaluation tasks and 3825 executable environments from 24 open-source repositories in the first version of our benchmark. Empirical evaluation reveals that the state-of-the-art agentic model, such as Claude 4.5 Opus, which achieves a 74.4% resolved rate on SWE-bench, succeeds on only 11.0% of tasks, opening new opportunities for advancing agentic coding. Moreover, benefiting from our automated task collection toolkit, FeatureBench can be easily scaled and updated over time to mitigate data leakage. The inherent verifiability of constructed environments also makes our method potentially valuable for agent training.

## 1 Introduction

Software development is rapidly evolving with the advent of large language models (LLMs) (Sapkota et al., 2025), marking a shift toward end-to-end agentic coding systems (Wang et al., 2025a). Recent advances, such as Claude Code (Anthropic, 2025b) and Qwen Code (Qwen, 2025) exemplify this evolution by introducing requirement-driven agents that autonomously plan, execute, and interact with external tools (*e.g.*, compilers) to iteratively tackle complex software development tasks (Gong et al., 2025), thereby relegating human intervention to a supervisory role.

Recently, various benchmarks have been introduced to assess this paradigm shift, including SWE-bench (Jimenez et al., 2024), PaperBench (Starace et al., 2025), and GitTaskBench (Ni et al., 2025). While these benchmarks have made significant contributions to task-oriented agentic coding, they are limited either by the narrow focus on bug-level scenarios or by reliance on handcrafted generation pipelines. As agentic coding expands toward more complex settings, such as feature-level

---

*Equal contribution.
†Corresponding Author.

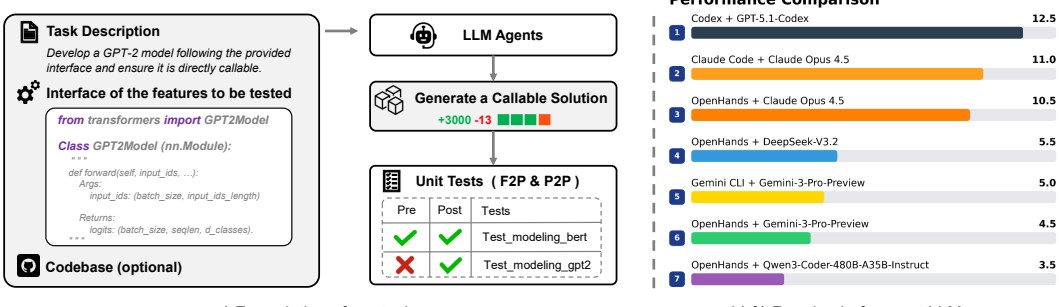

Figure 1: a) The agent must implement a directly callable feature based on the task description and interface definitions, either by developing from scratch or extending an existing repository. b) Our benchmark shows that even Claude Opus 4.5 achieves only a 11.0% solution rate.

development, these constraints hinder their ability to fully capture the capabilities of frontier code agents. Therefore, there is a need to build a challenging benchmark that broadens evaluation scope to feature-level scenarios, supported by automated collection toolkits to facilitate its future usage.

Constructing such a benchmark poses nontrivial challenges. Effective and execution-based evaluation of feature-level agentic coding generally depends on clearly defined functional interfaces to resolve ambiguities between the implementation and test criteria. However, these specifications are often absent in previous benchmarks. Furthermore, creating an automated data collection toolkit to support the scaling of benchmarks introduces additional complexities. Conventional pull request (PR)-based methods (Jimenez et al., 2024; Pan et al., 2025; Jain et al., 2025b) are ineffective in capturing complete feature patches, as these often span multiple PRs scattered across the timeline, making them difficult to associate. Moreover, many PRs lack tagging, hindering the reliable identification of feature contributions. Notably, PR-driven methods are inherently tied to the historical trajectory of commit submissions, limiting the tasks to fixed development combinations.

Motivated by these shortcomings, we introduce FeatureBench , a challenging benchmark that targets feature-oriented agentic coding scenarios. It integrates an execution-based evaluation pipeline and a test-driven toolkit for automatically collecting instances from Python repositories. As shown in Table 1, our bench provides the following characteristics:

1. **Feature-oriented real-world software development.** Unlike SWE-bench, which is dominated by bug-fixing issues with only about 18–22% of its instances corresponding to feature requests, our benchmark is explicitly designed to target systematic feature-level agentic coding. As shown in Figure 1, given human-like clear requirements (*e.g.*, interface signatures and high-level functional descriptions), our task entails the implementation of new capabilities either within an existing codebase or as standalone modules. For example, adapting the Transformers library (Wolf et al., 2020) for compatibility with Qwen3 (Yang et al., 2025a) or engineering FlashAttention (Dao et al., 2022) from scratch.

2. **Reliable execution-based evaluation.** Highly ambiguous requirements without explicit function signatures often introduce multiple valid implementations that are incompatible with the interface expected by unit tests. This misalignment complicates execution-based evaluation and typically necessitates additional manual inspection or LLM-based judgement (Starace et al., 2025; Seo et al., 2025). To mitigate this issue, we adopt a test-driven formulation strategy when constructing requirements. Each prompt explicitly specifies the clear interface definitions, import paths, and the descriptions of expected behaviors, and enforces that the solution must be directly callable, as illustrated in Figure 1. This method guarantees that a correct implementation will pass all associated tests, thereby enabling automated execution-based evaluation.

3. **Scalable instance collection toolkit.** To support the extensible creation of feature-oriented, realistic evaluation environments with fail-to-pass (F2P) and pass-to-pass (P2P) tests, as introduced in SWE-bench, we develop an automated generation pipeline driven by unit tests. The pipeline begins by selecting and executing F2P and P2P tests, followed by the construction of a dependency graph through dynamic tracing. Based on the traced dependencies, the system automatically extracts the implementation of the targeted features while ensuring the integrity of other features. The final problem statements are then synthesized. This approach enables us to generate naturally

| Benchmark | Feature-oriented Agentic Coding | Execution-based Evaluation | Scalable Instance Collection | Continually Updatable | Instance Number |
|---|---|---|---|---|---|
| BigCodeBench (Zhuo et al., 2025) | | ✓ | | | 1140 |
| LiveCodeBench (Jain et al., 2025a) | | ✓ | | | 454 |
| FullStackBench (Cheng et al., 2024) | | ✓ | | | 3374 |
| SWE-bench (Jimenez et al., 2024) | | ✓ | ✓ | ✓ | 500 |
| PaperBench (Starace et al., 2025) | ✓ | | | | 20 |
| Paper2Coder (Seo et al., 2025) | ✓ | | | | 90 |
| MLEBench (Chan et al., 2025) | ✓ | ✓ | | | 72 |
| DevEval (Li et al., 2025) | ✓ | ✓ | | | 20 |
| GitTaskBench (Ni et al., 2025) | ✓ | ✓ | | | 54 |
| FeatureBench (ours) | ✓ | ✓ | ✓ | ✓ | 200 |

Table 1: A comparison of FeatureBench with current coding benchmarks reveals that our bench emphasizes feature-level realistic software development. It leverages an execution-based evaluation pipeline and integrates a test-driven toolkit for the automatic generation of task instances.

    verifiable environments from any Python repository in a scalable and flexible manner, free from the constraints of the availability and predefined trajectory of human-written PRs or commits.

**4. Continually updatable.** Building on our collection toolkit, FeatureBench supports a continual supply of new task instances, enabling evaluation on tasks created after their training date, thus mitigating the risk of contamination. Using this pipeline, we have curated a benchmark with 200 evaluation instances and 3825 verifiable environments, created from May 2022 to September 2025, sourced from 24 real-world GitHub repositories in the first version of our benchmark.

We evaluate multiple state-of-the-art LMs on FeatureBench and find that they fail to solve all except the simplest tasks. Using the Codex agent framewor, GPT-5.1-Codex (medium reasoning) successfully completes 12.5% of the task cases. Furthermore, we carried out comprehensive experiments, offering insights into potential improvement directions on our benchmark.

In a nutshell, our contributions are three-fold: 1) We introduce FeatureBench, a benchmark for agentic coding that evaluates LLMs on solving feature-level, real-world complex tasks through an automated, execution-based evaluation pipeline. 2) We release a scalable, test-driven toolkit for instance collection that integrates seamlessly with our benchmark and automatically generates verifiable environments from Python repositories. Using this toolkit, we construct a benchmark comprising 200 evaluation tasks and 3825 executable environments from 24 open source GitHub repositories. 3) We benchmark state-of-the-art LLMs, including both open- and closed-source variants, and perform in-depth analysis to identify and highlight remaining challenges.

## 2 RELATED WORK

**Agentic Coding Benchmarks.** The most widely adopted benchmark for agentic coding is SWE-bench (Jimenez et al., 2024), whose verified subset has emerged as a standard for assessing LLMs. Although originally highly challenging, its success rate has increased from below 10% to over 70% within a year, reflecting rapid advances in LLM-based agents (Anthropic, 2025b; Yang et al., 2025a). Despite its importance, SWE-bench has notable drawbacks. It mainly focuses on bug fixing, with comparatively limited coverage of feature development tasks, which often span multiple PRs. Other benchmarks address narrower domains or predefined workflows. PaperBench (Starace et al., 2025) and MLE-Bench (Chan et al., 2025) focus on machine learning problems but rely on expert curation or high-quality cases from Kaggle. GitTaskBench (Ni et al., 2025) broadens task coverage but offers only 54 expert-designed tasks, while DevEval (Li et al., 2025) spans the development lifecycle but enforces fixed workflows with 22 handcrafted tasks. To tackle the above problems, we propose a challenging benchmark specifically designed for feature-oriented agentic coding scenarios. This benchmark integrates an execution-based evaluation pipeline and an automated toolkit that collects instances from Python repositories in a scalable manner.

**Scalable Collection Pipeline.** A verifiable environment is crucial for achieving better agentic coding. SWE-Gym (Pan et al., 2025) follows the pull-request based approach of SWE-bench, whereas R2E-Gym (Jain et al., 2025b) derives tasks from commits by synthesizing tests and back-translating code changes into problem statements with LLMs. These approaches mitigate scalability concerns but provide limited guarantees of evaluation quality. SWE-Smith (Yang et al., 2025b) synthesizes

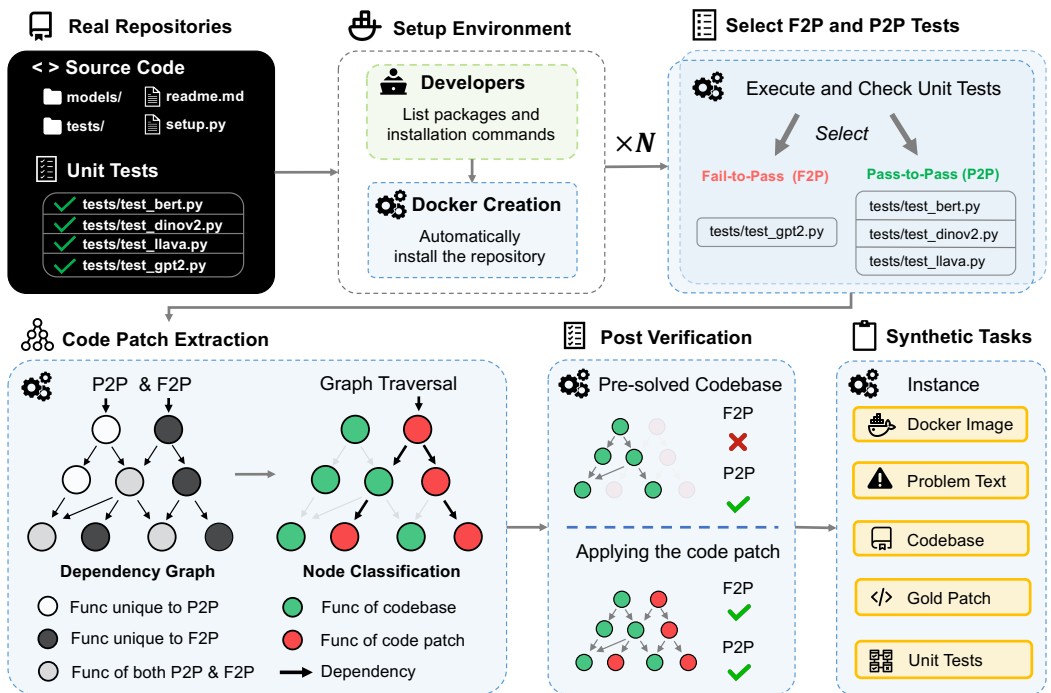

Figure 2: Given a GitHub repository, our automated toolkit initializes the development environment via Docker. For each benchmark instance, it validates and selects *fail-to-pass* and *pass-to-pass* tests. Then, the system performs dynamic tracing to capture runtime behavior and construct an object dependency graph. Leveraging this graph, the toolkit synthesizes code patches, derives corresponding pre-solved codebases, and formulates final problem statements. This pipeline has yielded 200 benchmark tasks and 3825 executable environments from 24 GitHub repositories.

tasks from repositories using heuristics such as LLM generation, procedural modifications, or pull-request inversion. SWE-Flow (Zhang et al., 2025) synthesizes data based on fail-to-pass tests but neglects pass-to-pass tests and does not ensure the proper functioning of other features in undeveloped codebases, resulting in discrepancies compared to actual development settings. Although successful, none of them can generate tasks that are both feature-oriented and reflective of real-world development scenarios. Our benchmark addresses these gaps by providing a test-driven, scalable tool for generating feature-level agentic coding tasks, complemented by a rigorous post-verification that ensures the integrity of undeveloped codebases, consistent with real-world scenarios.

## 3   FEATUREBENCH

FeatureBench establishes a benchmark for evaluating the capabilities of code agents in end-to-end software development tasks. The benchmark requires agents to interpret high-level goals and their associated code interfaces, autonomously manage execution environments, and synthesize correct and callable implementations either within existing codebases or as standalone solutions. Constructed with minimal human intervention, the benchmark leverages an automated pipeline that derives feature-oriented coding tasks from open-source repositories, thereby extending the scope of agentic coding beyond bug fixing to encompass feature development.

### 3.1   FEATURE-ORIENTED AGENTIC CODING

**Task Formulation.** As illustrated in Figure 1, each instance in FeatureBench provides the agent with a comprehensive problem statement. This includes a high-level task description, a specified functional interface, a blacklist of prohibited URLs to mitigate potential cheating of agents, and a dockerfile defining the execution environment. The agent is then tasked with generating a solution that addresses the problem, whether by editing existing code or implementing from scratch. Notably,

to facilitate automated and unambiguous evaluation, the agent's output is required to be a directly callable module. Its invocation path, function signature, including input and output variables as well as comprehensive annotations, are all explicitly provided within the problem statements.

**Difficulty.** In realistic settings, software development may proceed either by extending an existing codebase or by implementing a feature entirely from scratch. FeatureBench reflects these two scenarios with two difficulty levels. Level 1 ($L_1$) consists of incremental development within an existing repository based on task requirements, while Level 2 ($L_2$) requires constructing the same functionality from scratch.

**Metric Design.** Our evaluation protocol follows the established setup of SWE-bench (Jimenez et al., 2024), where each agent-generated solution is validated by executing its associated *fail-to-pass* (F2P) and *pass-to-pass* (P2P) tests. A task is considered resolved when the proposed solution successfully passes all these tests. We report three primary metrics: (1) *Resolved Rate*, the proportion of tasks fully solved, like SWE-bench; (2) *Passed Rate*, the average fraction of fail-to-pass tests passed per task, serving as a soft indicator of partial correctness; (3) *Token IO*, the average number of input and output tokens consumed, reflecting the computational efficiency of the agent.

## 3.2 BENCHMARK COLLECTION

**Execution Environment Configuration.** To rapidly set up an environment for a given repository, we manually specify installation commands (taking approximately three minutes), rather than relying on the more error-prone and uncontrollable approach of having the agent search for installation methods itself. Automated scripts are then used to configure the environment and package the repository into a Docker image. The benchmark includes 24 widely downloaded PyPI packages across various domains, such as visualization libraries and LLM infrastructure. Notably, human intervention is required only for this step of the pipeline, and the total human labor required to complete this for all 24 repositories amounts to less than one hour.

**Constructing *Fail-to-pass* and *Pass-to-pass* Tests.** We construct benchmark instances by identifying candidate test files in the repository using pytest's collection function, followed by validation through execution. For each instance, $n$ validated test files are designated as *fail-to-pass* (F2P) tests, as introduced in SWE-bench. These tests fail in the undeveloped repository but succeed once the agent correctly implements the target functionality. To additionally assess incremental development capability, we include $m$ randomly sampled validated files as *pass-to-pass* (P2P) tests, which are expected to pass both before and after the agent's solution. Since a single test file typically corresponds to one functional implementation, $n$ is usually set to one in our setting.

**Test-Driven Code Patch Extraction.** Obtaining the pre-solved codebase together with the corresponding code patch requires isolating the functionality linked to the F2P tests. However, the inherent ambiguity of functional boundaries in real-world codebases poses a significant challenge. Naively extracting relevant code fragments risks inadvertently disrupting other well-established features. As depicted in Figure 2, our approach mitigates this issue by incorporating P2P tests to accurately identify code modules required by other functions or those serving as foundational components of the repository. The detailed implementation is as follows:

- *Construct the object dependency graph.* We initiate the process by executing the available F2P and P2P test cases for a given benchmark instance. During runtime, we employ Python's built-in tracing facility to capture function call events and their dependencies. From this trace, we construct an object dependency graph in which each node represents a function and is enriched with metadata, including a unique identifier, source location, a list of dependent functions, and a binary flag indicating if the function was triggered during P2P tests.

- *Graph traversal and node classification.* To distinguish functional components, a large language model analyzes the F2P test files and separates the imported functions related to the target feature from those that serve supporting roles in the testing process. The nodes identified as central to the undeveloped feature serve as the initial entry points for a breadth-first traversal of the graph. During this traversal, nodes are systematically classified: those encountered in P2P executions are designated as *remained*, while nodes not observed in P2P runs are classified as *extracted*.

- *Extracting the code.* The traversal process yields a subset of graph nodes identified as relevant to the intended functionality. In the final stage, the corresponding segments of source code are

| Scaffold | Model | Lite | | | Full | | |
|---|---|---|---|---|---|---|---|
| | | % Passed | % Resolved | # Token I/O | % Passed | % Resolved | # Token I/O |
| OpenHands | Qwen3-Coder-480B-A35B-Instruct | 38.31 | 6.7 | 2.6M / 16k | 24.55 | 3.5 | 2.0M / 14k |
| OpenHands | DeepSeek-V3.2 | 35.94 | 6.7 | 3.1M / 24k | 26.30 | 5.5 | 3.1M / 23k |
| OpenHands | Gemini-3-Pro-Preview[†] | 45.14 | 10.0 | 6.0M / 41k | 30.08 | 4.5 | 6.2M / 40k |
| Gemini-CLI | Gemini-3-Pro-Preview[†] | 43.38 | 10.0 | 2.6M / 13k | 32.43 | 5.0 | 2.5M / 12k |
| Claude Code | Claude Opus 4.5 | 59.12 | 20.0 | 9.0M / 35k | 43.29 | 11.0 | 7.5M / 34k |
| Codex | GPT-5.1-Codex[‡] | 60.22 | 20.0 | 6.6M / 39k | 41.66 | 12.5 | 6.3M / 39k |
| OpenHands | Claude Opus 4.5 | 67.18 | 20.0 | 8.8M / 29k | 45.53 | 10.5 | 8.1M / 29k |

Table 2: The performance of various frontier large models combined with advanced agentic frameworks on the Lite and Full evaluation sets of our benchmark. Models marked with [†] use low reasoning, and [‡] use medium reasoning.

extracted from the original codebase. This operation produces a modified codebase devoid of the target functionality and a complementary code snippet that realizes the previously absent feature.

**Post Verification.** To ensure the successful extraction of the target functionality from the codebase without affecting other components, we implement a rigorous verification process. The first step involves validating the pre-modified codebase by ensuring that it passes all P2P tests, thereby confirming its integrity. Simultaneously, it must fail all F2P tests, demonstrating that the target functionality has been effectively removed. Following this, we assess the accessibility of all utility functions required for the F2P tests in the modified codebase. This step ensures that the changes made are confined to the target functionality and do not inadvertently impact other core dependencies. Finally, reapplying the patch to the undeveloped codebase should allow all tests to pass, confirming the patch's correctness.

**Problem Statement Generation.** By leveraging the extracted code snippet, the pre-modified codebase, and the corresponding unit tests, we automatically generate the problem statement for each instance. This procedure includes the derivation of the feature signatures, which encompass the types of input and output variables, alongside the functional description as inferred from the code docstrings. In the absence of such docstrings, we employ a large language model to generate them directly from the code snippet. Further details can be found in the appendix.

To this end, our pipeline automatically generates the core components of each instance: a natural language problem statement, an undeveloped codebase, a verified code patch, and a suite of unit tests corresponding to required features. The sole manual intervention required is the specification of the repository's installation procedure, a process that takes approximately three minutes per repository.

### 3.3 BENCHMARK CONFIGURATION

**Full Set.** Leveraging our pipelines, we configured the number of P2P test files to five and curated 3825 coding environments derived from 24 Python repositories. To ensure the benchmark meaningfully challenges best-performing agents, we restricted inclusion to tasks exceeding 100 lines of pending implementation, encompassing at least 10 F2P test points, with test files initially committed after May 2022. This filtering yielded 200 high-quality instances comprising the full set.

**Lite Set.** Evaluating LMs on our bench can be time-consuming and, depending on the model, require a costly amount of compute or API credits, as illustrated in Table 2, where the average number of input tokens approaches the million-token mark. To facilitate wider adoption of FeatureBench , we randomly selected 30 instances from the full set to create a streamlined lite set.

## 4 EXPERIMENTS

### 4.1 PERFORMANCE ON FEATUREBENCH

#### 4.1.1 BASELINE

To establish strong baselines, we adopt the OpenHands (Wang et al.) framework for software development agents, which tops the SWE-bench. In the experiments, the maximum of steps per task is

|  |  | SWE-bench | Ours |
|---|---|---|---|
| Problem Texts | Length (Words) | 195.1 | 4818.0 |
| Gold Solution | # Lines | 32.8 | 790.2 |
|  | # Files | 1.7 | 15.7 |
|  | # Functions | 3 | 29.2 |
| Tests | # Fail to pass (test points) | 9.1 | 62.7 |
|  | # Total (test points) | 120.8 | 302.0 |

Table 3: Average numbers characterizing different attributes of a SWE-bench task instance, as well as our FeatureBench ($L_1$ set).

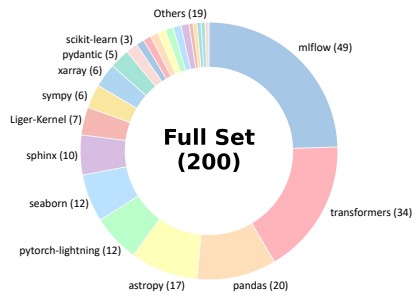

Figure 3: Distribution of our benchmark across 24 GitHub repositories.

| Model | SWE-Bench Verified | | FeatureBench subset | | |
|---|---|---|---|---|---|
|  | % Resolved | | % Passed | % Resolved | # Token I/O |
|  | mini-SWE-agent | OpenHands |  | OpenHands |  |
| DeepSeek-V3.2 | 60.00 | - | 22.98 | 0.0 | 3.8M / 25k |
| Qwen3-Coder-480B-A35B-Instruct | 55.40 | 69.60 | 23.46 | 0.0 | 2.3M / 16k |
| Gemini-3-Pro-Preview | 74.20 | - | 30.05 | 0.0 | 6.7M / 45k |
| Claude Opus 4.5 | 74.40 | - | 41.08 | 5.2 | 9.7M / 34k |

Table 4: Compare the performance of the frontier agents on SWE-bench and our FeatureBench, using a subset of our benchmark with repositories shared with SWE-bench for a fair comparison.

set as 500 by default. Internet access is freely available, while no specific browser-use tools are provided. To ensure the integrity of our evaluation, robust anti-cheating mechanisms are incorporated to prevent agents from assessing the ground-truth repositories (see the appendix for details).

We evaluate seven *scaffold+model* configurations with frontier LLMs, including DeepSeek-V3.2 (DeepSeek, 2025), Qwen3-Coder-480B-A35B-Instruct (Qwen Team, 2025), Gemini-3-Pro-Preview (low reasoning) (Google, 2025a), Claude Opus 4.5 (Anthropic, 2025a), and GPT-5.1-Codex (medium reasoning) (OpenAI, 2025b) under representative agentic scaffolds (OpenHands (Wang et al., 2025b), Gemini-CLI (Google, 2025b), Claude Code (Anthropic, 2025b), and Codex (OpenAI, 2025a)). The results are presented in Table 2. As can be seen, even the most capable settings, *i.e.*, Claude Code (routing) + Claude Opus 4.5 and Codex + GPT-5.1-Codex (medium reasoning), resolve only 11.0% and 12.5% of the tasks on the *Full* set, respectively. This underscores the highly challenging nature of the feature-oriented development tasks in our FeatureBench, which require agents to write substantial amounts of code and pass comprehensive test suites.

For a more nuanced evaluation, we further analyze passed rates and token consumption by different LLMs. The passed rates, while remaining at a low level of below 50%, are much higher than the resolved rates. This discrepancy indicates that current agents often produce seemingly plausible solutions with a large underlying gap from truly solving the problem, which accounts for the common need of tedious debugging for AI-generated code. Regarding token consumption, all LLMs consume over one million input tokens. Given the low resolved rates, this reflects the extremely low efficiency of existing agents in tackling real-world development tasks, which is thus an important topic for future research. In addition, a high consistency is observed in the rankings of different LLMs across the *Lite* and *Full* sets in terms of both pass and resolved rates, demonstrating the representativeness of the Lite set.

### 4.1.2 COMPARISON WITH SWE-BENCH

Compared with the SWE-bench (Jimenez et al., 2024), our FeatureBench introduces a more challenging suite of development tasks. It encompasses 16 additional popular repositories apart from 8 repositories originally covered by the SWE-bench, the full list of which is shown in Figure 3. Table 3 presents comparative statistics illustrating the task difficulties across the two benchmarks. Specifically, the tasks in our benchmark exhibit a substantial increase of complexity in terms of the length of problem texts, number of lines, files and functions to be edited as well as the number of tests to pass. These enhancements necessitate agents with strong long-context understanding and management capabilities alongside comprehensive problem analysis to handle diverse test cases.

| | Models | % Resolved | % Passed |
|---|---|---|---|
| Original | Gemini-3-Pro-Preview[†] | 10.0 | 42.4 |
| | GPT-5.1-Codex[‡] | 16.7 | 53.9 |
| Verified | Gemini-3-Pro-Preview[†] | 10.0 | 43.4 |
| | GPT-5.1-Codex[‡] | 20.0 | 60.2 |

Table 5: An ablation study to evaluate the necessity of manual verification for the examples generated by our system. Models marked with [†] use low reasoning, and [‡] use medium reasoning.

| Models | Steps | % Resolved | % Passed |
|---|---|---|---|
| Gemini-3-Pro-Preview[†] | 50 | 6.7 | 22.9 |
| | 100 | 6.7 | 43.8 |
| | 500 | 10.0 | 45.1 |
| Qwen3-Coder-480B-A35B-Instruct | 50 | 3.3 | 28.9 |
| | 100 | 3.3 | 30.4 |
| | 500 | 6.7 | 38.3 |

Table 6: An ablation study on the max execution steps of OpenHands with Gemini-3-Pro-Preview and Qwen3-Coder-480B-A35B-Instruct in Lite Set. Models marked with [†] use low reasoning.

| Model | Without Interface | | | Visible Unit Tests | | |
|---|---|---|---|---|---|---|
| | % Resolved | % Passed | # Token I/O | % Resolved | % Passed | # Token I/O |
| Gemini-3-Pro-Preview[†] | 3.3 (-6.7) | 25.3 (-18.1) | 7.0M / 10K | 60.0 (+50.0) | 80.6 (+37.2) | 6.9M / 18K |
| GPT-5.1-Codex[‡] | 16.7 (-3.3) | 42.0 (-18.2) | 7.6M / 38K | 63.3 (+43.3) | 80.9 (+20.7) | 8.2M / 46K |

Table 7: Performance comparison of lite set with visible unit tests and without interface. Models marked with [†] use low reasoning, and [‡] use medium reasoning.

For a more grounded analysis, we further compare the performance of agents on the SWE-bench and our FeatureBench . To draw a more aligned comparison, we construct a subset of our benchmark including only repositories shared with SWE-bench. The results in Table 4 reveals a stark performance gap between the two benchmarks in terms of resolved rate. Specifically, the most capable Claude Opus 4.5 only resolves 5.2% of the tasks in our FeatureBench subset in contrast to the 74.40% on the SWE-bench. This indicates the highly challenging nature of our benchmark, which provides considerable room for future improvement and establishes a rigorous testbed to measure the upper bound of existing agents.

### 4.1.3 FAILURE CASES ANALYSIS

We conduct a failure case analysis based on the results in our full set from the Claude Opus 4.5 model, leading to the following findings.

**Limitations in Code Reasoning.** As shown in Figure 4, the dominance of `NameError` suggests that current LLMs still struggle with cross-file dependency resolution. When a feature spans multiple files, models often focus on local edits without consistently re-establishing all necessary references, leading to unresolved symbols and frequent name-related failures. This highlights a key limitation in maintaining coherent program context beyond a single file.

**The "Idle Habits" of LLMs.** We also find that current LLMs exhibit a tendency toward "laziness". For example, they often resort to guessing (even hallucinating) the interface or attributes of components defined across files, rather than performing the actual file reading required to retrieve precise prototypes and members. This behavior leads to a considerable number of both `TypeError` and `AttributeError` occurrences.

**Appropriate Information in FeatureBench.** Among the remaining failures, `AssertionError` becomes the most frequent category. This suggests that a substantial portion of LLM-generated solutions can run to the assertion checkpoints without earlier runtime crashes. This result underscores that FeatureBench can effectively provide the LLMs with appropriate information to generate complete programs.

### 4.2 ABLATION STUDY

#### 4.2.1 ANALYZING THE QUALITY AND NECESSITY OF OUR BENCHMARK DESIGN.

**Without Interface.** We performed an ablation study to assess the role of explicit interface specification in agent performance. For controlled comparison, we employed the lite set, systematically removing function signatures and call path annotations from the prompts. As shown in Table 7, this

| Difficulty | Scaffold | Models | % Resolved | % Passed |
|---|---|---|---|---|
| $L_1$ | OpenHands | Qwen3-Coder-480B-A35B-Instruct | 3.6 | 22.4 |
| | OpenHands | DeepSeek-V3.2 | 4.8 | 20.8 |
| | OpenHands | Claude Opus 4.5 | 11.4 | 46.2 |
| | OpenHands | Gemini 3 Pro[†] | 4.2 | 29.0 |
| | Gemini-CLI | Gemini 3 Pro[†] | 4.8 | 32.1 |
| | Codex | GPT-5.1-Codex[‡] | 13.9 | 43.0 |
| | Claude Code | Claude Opus 4.5 | 11.4 | 43.6 |
| $L_2$ | OpenHands | Qwen3-Coder-480B-A35B-Instruct | 2.9 | 35.2 |
| | OpenHands | DeepSeek-V3.2 | 5.9 | 32.6 |
| | OpenHands | Claude Opus 4.5 | 5.9 | 42.2 |
| | OpenHands | Gemini 3 Pro[†] | 5.9 | 35.6 |
| | Gemini-CLI | Gemini 3 Pro[†] | 5.9 | 34.0 |
| | Codex | GPT-5.1-Codex[‡] | 5.9 | 35.2 |
| | Claude Code | Claude Opus 4.5 | 8.8 | 41.9 |

Table 8: Performance comparison of tasks with different difficulty levels in FeatureBench. Models marked with [†] use low reasoning, and [‡] use medium reasoning.

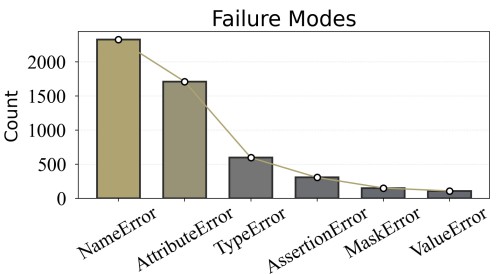

Figure 4: Failure modes of the Claude Opus 4.5. Models marked with [†] use low reasoning, and [‡] use medium reasoning.

removal leads to a marked decline in task success rates. The results confirm that clearly defined interfaces are critical for enabling effective reasoning and program synthesis by LLM-based agents.

**Sample Quality.** Our automated data generation pipeline yields high-quality, evaluation-ready samples with minimal human intervention, supported by a rigorous post-verification process. To assess the fidelity of these samples, we conducted an ablation study in which a senior engineer with five years of industry experience in AI infrastructure and system architecture independently revised the prompts in the lite set. The verification details are provided in Appendix Figures 20 and 21. As shown in Table 5, model performance on the manually revised subset is highly consistent with the original dataset. These results affirm the reliability and robustness of our automated data pipeline.

**Lines of Code and Task Initial Commit Date.** Figure 5 explores the relationship between task pass rates, initial commit timestamps, and the number of lines of code required for task completion. We observe a clear negative correlation between pass rate and code length, indicating that tasks involving more lines of code are inherently more challenging for current large models. In contrast, task performance shows minimal dependence on commit time, likely because the task set remains largely unexplored by existing models. To further understand why commit time has little influence, we analyze how feature complexity evolves over time. Specifically, the lower panel of Figure 5 plots the normalized trends of code length and pass rate across commit periods. The two normalized curves exhibit highly similar fluctuations, reinforcing that variation in task performance is driven far more by feature complexity than by commit time. However, as agentic systems increasingly participate in feature development

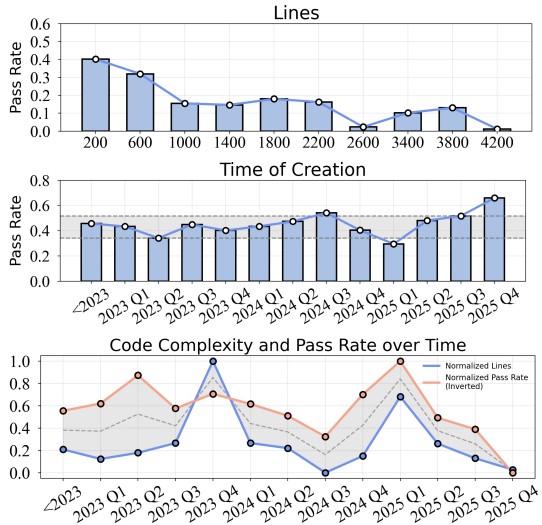

Figure 5: The pass rate of Claude Opus 4.5 in our benchmark varies with the number of code lines and task creation time.

workflows, the risk of data leakage may become more pronounced and should be monitored in future benchmark design.

**Comparison between $L_1$ and $L_2$ Subset.** Comparison between $L_1$ and $L_2$ Subsets. Our benchmark defines two evaluation settings: $L_1$, where new functionalities are incrementally added to an existing codebase, and $L_2$, where functionalities are implemented entirely from scratch. All conditions are held constant across both settings, except for the presence or absence of initial code context.

This distinction leads to notably different levels of reasoning complexity. In the $L_1$ setting, the agent still has access to most of the original codebase except for the functions and classes removed along the traced execution path. This partial repository shows how the feature fits into the surrounding code and gives the agent contextual clues about expected behavior. As a result, $L_1$ tasks are more guided, since only the missing implementations need to be completed. In contrast, $L_2$ tasks remove

| Metric | Precision | Recall | F1 Score | Accuracy |
|--------|-----------|--------|----------|----------|
| Value | 81.03% | 89.24% | 84.94% | 91.74% |

Table 9: Performance of the LLM classifier for identifying top-level tested objects.

all surrounding code. The agent does not see any part of the original repository and must rely only on the interface to implement the required functionality. Without the structure provided by the existing codebase, the agent has to reconstruct the full logic and organization of the feature entirely from scratch, which makes $L_2$ substantially more difficult. As shown in Table 8, the from-scratch ($L_2$) setting is more challenging, with lower resolved rates: performance on $L_2$ varies little across settings, suggesting that removing the codebase structure creates a common bottleneck that hampers coherent multi-step reasoning and end-to-end implementation.

**Accuracy of LLM-based Top Import Classification.**

To validate the reliability of our LLM-based classifier for identifying top-level tested objects in test file, we conducted a quantitative evaluation against expert annotations. Domain experts evaluated all 605 import statements in the Lite Set and identified 158 of them as top-level tested objects. The details of the procedure are provided in Appendix Figure 19. Table 9 reports the performance of the LLM classifier. These results indicate that LLMs can accurately identify tested objects at scale, supporting the use of LLM-based classification in our data construction pipeline.

### 4.2.2 ANALYZING THE KEY FACTORS IN BUILDING END-TO-END CODEAGENTS

**Visible Unit Tests.** We conducted an ablation study to assess the impact of providing accurate unit tests on agent performance in complex coding tasks. In this setting, the agent was given access to ground-truth unit tests alongside the Lite set. As shown in Table 7, both task success rates and pass rates increased significantly. These findings underscore the importance of high-quality unit test generation as a key factor in enabling robust agentic coding.

**Longer Execution Steps.** Table 6 reports the effect of increasing the maximum number of execution steps on model performance. Increasing the maximum step size from 50 to 100 results in notable performance gains for both Gemini-3-Pro-Preview and Qwen3-Coder-480B-A35B-Instruct. However, beyond this threshold, the improvements become marginal.

## 5 CONCLUSION

In this work, we introduce FeatureBench , a novel benchmark designed to evaluate the capabilities of LLM-powered agents in realistic, feature-oriented software development scenarios. Leveraging test-driven task extraction and execution-based evaluation, FeatureBench overcomes key limitations of existing benchmarks by enabling greater task diversity, scalability, and verifiability. Empirical results reveal that current agentic systems face persistent challenges in planning, reasoning, and managing long-horizon tasks. With its extensible and automated design, FeatureBench offers not only a rigorous evaluation framework but also a foundation for the development of next-generation agentic coding models.

## ACKNOWLEDGEMENTS

This work was supported in part by the Beijing Natural Science Foundation (No. L257015 and No. L257004) and in part by the National Natural Science Foundation of China (No. 62320106010).

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

APPENDIX

# A    DETAILED BENCHMARK COLLECTION

This section complements the details of benchmark construction (Sec. 3.2), which contains detailed recipes of the data collection, patch extraction, and prompt design, along with a fuller characterization of the task instances.

## A.1    DATA COLLECTION PIPELINE

**Environment Setup.**

For each selected repository, we manually prepare an environment configuration file (see Figure 6 for an example). Empirical observations indicate this procedure can be accomplished within three minutes. Upon completion of environment configuration, our pipeline constructs a Docker image, with all subsequent operations executed within this sandboxed environment. This is the sole stage requiring human intervention. All succeeding stages operate under full automation.

**Patch Extraction.** The patch extraction process consists of four main steps.

*Patch Extraction Step 1: Dependency Graph Construction.* This procedure generates function-level dependency graphs for all test files within the code repository, establishing the foundation for subsequent patch extraction operations. We leverage pytest's intrinsic test case collection mechanism to aggregate all viable test cases at the file granularity, where each file contains a potential test case. For each test case, we execute the test within the sandbox environment, selecting test cases that achieve complete success as *fail-to-pass* (F2P) instances. Concurrent with test execution, we construct function-level dependency graphs for each F2P instance utilizing a dynamic tracing library.

*Patch Extraction Step 2: LLM Classification.* For each F2P test file, we employ an LLM to differentiate between imported objects serving as test targets versus those functioning as test dependencies and general utilities. We provide the LLM with the test file's name and content as classification references. Our prompt template for the LLM to classify is illustrated in Figure 9. Objects classified through this methodology are designated as top-level objects, representing directly imported interfaces by the test file.

*Patch Extraction Step 3: Pass-to-pass (P2P) Selection.* For each F2P instance, we select multiple *pass-to-pass* cases. These P2P cases are executed after coding agents finishing implementations to ensure existing functionalities remain normal. Since the aforementioned top-level objects of F2P cases will be removed from codebases, here the pass-to-pass cases should not share top-level objects with the F2P cases. For this reason, if we find only a few P2P cases have different top-level objects from F2P cases, it may indicate erroneous classification of general utilities as top-level objects by the LLM. In this circumstance, we will reconsider the top-level objects according to their invocation frequency.

*Patch Extraction Step 4: Final Extraction.* For each F2P case, we utilize top-level objects as entry points and execute BFS according to the constructed dependency graph. Node objects belonging to P2P are designated as *remained*, while others are marked as *extracted*. Nodes marked as *extracted* are added to the BFS queue for continued traversal. BFS termination occurs upon queue finish or when extracted code lines reach our predetermined maximum value, randomly selected between 3000 and 5000 lines per case. Finally, we remove objects marked as *extracted* from the codebase, yielding a complete codebase with F2P functionality eliminated.

**Post-verification.** For each codebase after code patch extraction, we conduct post-verification to ensure the modified codebase has normal functionality. Specifically, we execute F2P within the modified codebase, expecting pass rates below a predetermined parameter. Then we further execute all selected P2P cases, ensuring complete test passage.

**Environment Configuration File part 1 of 2**

```
SPECS_LITGPT = {

    "repository": "Lightning-AI/litgpt",
    "commit": "22c2a4f",
    "clone_method": "https",
    "base_url": None,
    "base_image": "python310",
    "rebuild_base_image": False,
    "rebuild_instance_image": False,
    "custom_instance_image_build": [],
    "pre_install": [],
    "install": "pip install -e '.[extra]'",
    "pip_packages": [],
    "docker_specs": {
        "run_args": {
            "cuda_visible_devices": "0,1,2,3",
            "shm_size": None,
            "cap_add": [],
        },
        "custom_docker_args": []
    },

    "test_scanner_cmd": TEST_DISCOVERY_DEFAULT,
    "timeout_scanner": 300,
    "scan_cache": True,
    "start_time": None,
    "min_test_num": 1,
    "max_f2p_num": -1,
    "max_p2p_num": -1,
    …
```

Figure 6: Environment Configuration File part 1 of 2

**Environment Configuration File part 2 of 2**

```
…
    "test_cmd": TEST_PYTEST_VERBOSE,
    "timeout_run": 1200,
    "timeout_one": 10,
    "test_cache": True,

    "test_dynamic_cmd": TEST_DYNAMIC_TRACE_DEFAULT,
    "timeout_dynamic": 1200,
    "dynamic_cache": True,

    "llm_cache": True
    "batchsize_top": 5,
    "max_depth_top": 5,
    "min_p2p_files": 1,
    "max_p2p_files": 5,

    "p2p_cache": True,
    "max_code_line_lower_bound": 3000,
    "max_code_line_upper_bound": 5000,

    "data_cache": True,
    "timeout_collect": 300,
    "f2p_pass_rate_threshold": 0.3,
    "llm_prompt_for_case": True,

    "library_name": "litgpt",
    "black_links": [
        "https://github.com/Lightning-AI/litgpt/"
    ]
}
```

Figure 7: Environment Configuration File part 2 of 2

## A.2 DATA FORMAT AND PROMPT DESIGN

In this section, we present the essential components included in a qualified example (covering both $L_1$ and $L_2$ tasks), illustrating the test case format of our benchmark, organization of our prompts, and the effectiveness of using an LLM to supplement missing docstring entries.

**Directory Structure of Generated Instances.** Each successfully generated instance includes a directory structure containing four main files: `problem_statement.md`, `patch.diff`, `test_patch.diff` and `instance.json`. Specifically, `problem_statement.md` serves as the generated task prompt; `patch.diff` and `test_patch.diff` represent the gold patch and test patch, respectively; while `instance.json` records metadata such as the task ID, source repository, and commit ID.

**Prompt Structure and Organization.** For each successfully generated instance, we construct a tailored and detailed `problem_statement.md` file as input to the agent. The content of `problem_statement.md` consists of two components: *Task* and *Interface Descriptions*. All prompts are generated automatically without manual labor, following a unified prompt template combined with an instance-specific configuration file via scripting. As shown in Figure 12, the ***Task*** section provides the agent with an overview of the task under the heading "Core Functionality." "Main Features and Requirements" describes the essential code features and requirements. The mandatory components that must be implemented are outlined under "Key Challenges." Finally, the "NOTE" subsection provides specific requirements. In the ***Interface Descriptions*** section (as shown in Figure 13), we offer detailed instructions on how to construct the code. This includes requirements for file locations and structure, suggestions for implementing interfaces, and specific objectives related to the current task.

**Prompt Design: $L_1$ vs. $L_2$.** It is noteworthy that there are subtle differences between the $L_1$ and $L_2$ prompts. In the *Task* section of the prompt, for $L_2$ examples, we require the agent to independently implement the solution from scratch, without access to the repository's codebase. To enforce this constraint, the agent is instructed not to download the repository, and even in the event of its doing so, it is instructed not to install it. This is closely monitored during the testing phase, where we have set up mechanisms to check whether the agent engages in any unauthorized actions, such as cheating. Figure 15 and Figure 16 show a specific $L_1$ and $L_2$ prompt, respectively, and they both come from the test-layer-norm test in liger-kernel library.

**Supplementation for Missing Docstrings.** In cases where the source code lacks adequate documentation regarding function interfaces or behavior, we leverage a LLM to infer and complete the missing information. Figure 11 illustrates the prompt for docstring generation. The LLM-generated docstring is exemplified in Figure 18, where the LLM is used to supplement the missing docstring in the functional description.

## B DETAILED BENCHMARKING PROCESS

Our benchmarking pipeline is organized into two sequential stages: (1) agent inferring and (2) automated evaluation. The following subsections provide a detailed explanation of two phases.

### B.1 INFERRING

During the inference phase, after initializing the task environment image, we proceed as follows based on the setting:

For $L_1$: We use `patch.diff` to remove the target feature from the origin repository within the image and remove the corresponding F2P test file.

For $L_2$: We completely remove the entire code repository from the image.

Subsequently, the agent is deployed within the environment and provided with `problem_statement.md` as the prompt of the task. Upon completion, we extract the modifications made by the agent as a new `patch.diff` file for later evaluation.

## B.2 EVALUATION

During the evaluation phase, we utilize the same base image and reset the repository to its pre-inferring state (replicating the setup procedures for L1 and L2 described above). And then:

For $L_1$: We apply the agent-generated patch.diff to the repository and reintroduce the F2P test file.

For $L_2$: We apply the agent-generated patch.diff to the image and install it as the `agent_code` library. We then restore the original repository and apply the `test_patch.diff`.

Finally, we employ the pytest framework to perform automated testing and get the pytest report.

## B.3 METRICS

The evaluation process produces raw output from the `pytest` framework, which is subsequently processed to extract key statistics, including *total*, *passed*, *failed*, *skipped*, *error*, *xfail*, and *xpass*.

From these statistics, we derive two primary evaluation metrics: *pass_rate* and *is_solved*. The *pass_rate* is defined as the ratio of successfully passed tests to the total number of executed F2P tests. The binary metric *is_solved* indicates whether the task is fully solved. It is assigned a value of 1 if pytest command exit with 0 (both F2P and P2P), and assigned a value of 0 otherwise.

## B.4 CHEATING PREVENTION AND DETECTION

To protect against potential cheating attempts by agents, such as using `pip install <package>` followed by inspecting the source code of the installed package, we implement a twofold defense mechanism. First, defensive prompting is incorporated into the task descriptions to discourage such behavior. Second, the evaluation framework conducts an automated inspection of the agent's execution logs after task completion to identify suspicious activities.

The log inspection process searches for regular expression patterns that indicate unauthorized attempts to access the source code of installed packages. If any pattern matches, the agent is flagged for potentially dishonest behavior in the evaluation report.

```
r'"message".*cat /usr/local/lib/python\d+\.\d+'
r'"command".*cat /usr/local/lib/python\d+\.\d+'
r'"message".*reading file: /usr/local/lib/python\d+\.\d+'
r'"message".*reading /usr/local/lib/python\d+\.\d+'
```

These patterns capture attempts to directly access files within the Python library directory, which is classified as a form of cheating under the evaluation criteria.

## C ANALYSIS OF FALURES OF GEMINI 3 PRO MODEL

In our baseline experiments, we found that the Gemini 3 Pro model performed poorly. Through analysis of the model output logs, we discovered that this may be caused by a lack of strict adherence to JSON schemas during tool invocation. Specifically, we observed a recurrent pattern of parameter key hallucination, where the model substituted valid schema definitions with semantically similar but syntactically incorrect keys.

For example, in case `astropy__astropy.b0db0daa.test_basic_rgb.067e927c.lv1`, the model produced the output shown in Figure 8, the model attempted to invoke `read_file` using the argument `path` instead of the strictly defined `file_path`. This suggests that while the model understands the intent of the tool, it struggles to suppress its internal priors in favor of the provided API constraints.

This situation accounts for the majority of cases in our evaluation, This indicates that Gemini 3 Pro seems to have certain deficiencies in completing large-scale code editing tasks.

| Benchmark | Task Source | F2P/P2P | Real-world software development | Agent Eval. | Avg. LoC |
|---|---|---|---|---|---|
| SWE-Bench | PR | ✓ | ✓ | ✗ | 32.8 |
| SWE-Dev | Unit Tests | ✗ | ✗ | ✗ | 190 |
| FeatureBench | Unit Tests | ✓ | ✓ | ✓ | **790.2** |

Table 10: Comparison of FeatureBench with SWE-Bench and SWE-Dev.

| Benchmark | Full Implementation | Realistic Software Development | Scalability |
|---|---|---|---|
| commit0 | ✗ | ✗ | ✗ |
| FeatureBench | ✓ | ✓ | ✓ |

Table 11: Comparison of FeatureBench with SWE-Bench and commit0.

## D COMPARISON WITH EXISTING BENCHMARKS

This section provides additional comparisons between FeatureBench and two representative benchmarks, SWE-Dev (Du et al., 2025) and commit0 (Zhao et al., 2024). These comparisons complement the high-level discussion in Section 2 and clarify the distinctions in task sources, construction pipelines, evaluation settings, and scalability.

**Comparison with SWE-Dev** SWE-Dev derives tasks from unit tests and LLM-generated problem requirement descriptions (PRDs). Its task formulation and construction pipeline differ substantially from FeatureBench, particularly in how tasks are specified, validated, and filtered. Table 10 provides a concise comparison, followed by brief clarifications of the key distinctions.

- *Realistic development workflow and stricter construction.* FeatureBench preserves the original, well-developed features of each repository and leaves only the target feature unimplemented, closely matching incremental development. This is enforced through precise patch extraction, F2P/P2P filtering, and strict post-verification. SWE-Dev omits P2P verification and does not perform post-verification, allowing patches that unintentionally break existing behavior.

- *Interface-driven task specification with minimal ambiguity.* SWE-Dev uses LLM-generated PRDs, which naturally introduce ambiguity. FeatureBench instead exposes native top-level interfaces—function signatures and invocation paths extracted directly from the codebase—ensuring clear, deterministic, and implementable task specifications.

- *Agent-based evaluation in this work.* SWE-Dev reports results for LLM and multi-LLM settings but does not evaluate coding agents. FeatureBench conducts end-to-end agent experiments using a unified OpenHands scaffold with multiple LLM backends (Claude, GPT, Gemini, Qwen), providing a realistic assessment of agent performance.

- *More realistic and complex feature-level tasks.* SWE-Dev tasks involve roughly 190 LoC across three files. FeatureBench tasks require around 790.2 LoC across more files and substantially more test points, reflecting the multi-file, cross-module modifications typical in real feature development.

**Comparison with commit0** commit0 studies whether LLMs can reconstruct entire libraries from documentation and high-coverage test suites. This setup differs markedly from FeatureBench, whose focus is real-world feature development with full, from-scratch implementations and scalable construction. Table 11 summarizes the key distinctions, with brief explanations provided below.

- *Real-world development and full implementation.* In commit0, only the bodies of functions and classes are removed while definitions and architectural scaffolding remain, making tasks closer to fill-in-the-blank partial completions. FeatureBench removes definitions, imports, and associated logic, ensuring that the target feature is fully absent and must be implemented from scratch, better aligning with real development workflows.

- *Low-cost scalability to new repositories.* commit0 requires repositories with well-organized documentation and very high test coverage (>90%), severely limiting applicability. FeatureBench requires only a runnable unit-test suite; after a short configuration step, the rest of the pipeline is fully automated, enabling efficient scaling across a wide range of real-world codebases.

# E    DATASET OVERVIEW AND EXPERIMENTAL RESULTS

The construction of the dataset resulted in 200 evaluation tasks derived from 3825 candidate coding environments across 24 Python repositories. These repositories encompass a wide range of domains, including machine learning, scientific computing, visualization, web frameworks and fundamental software engineering utilities. This diversity ensures the dataset captures a broad spectrum of real-world coding scenarios.

To promote transparency and reproducibility, the appendix contains two tables that describe the dataset composition. The Table 13 provides an overview of each repository, including summary information and licensing details. The Table 12 presents quantitative statistics such as the average number of extracted code lines and the number of test points in the test suite.

To illustrate the dataset structure, we include an example of an individual data entry. Each entry includes the following fields: instance_id, patch, test_patch, FAIL_TO_PASS, PASS_TO_PASS, image_name, repo, base_commit, problem_statement, and repo_settings. The specific meaning of each field is detailed in Table 14.

Additionally, the experimental results, summarized in seven comprehensive tables (Table 15 to Table 21), evaluate the performance of multiple large language models across the dataset. Each table reports three repository-level average metrics: Passed, Resolved, and Token IO. These results provide insights into model capabilities, including task pass rates, resolution rates, and token input-output statistics. The results demonstrate the strengths and limitations of current coding agents in diverse scenarios, forming a foundation for future advancements in agentic coding research.

| Repo | # Lines | # Files | # Functions | # Test points |
|---|---|---|---|---|
| pytorch-lightning | 765.4 | 24.0 | 35.7 | 42.0 |
| metaflow | 520.0 | 1.0 | 5.0 | 31.0 |
| astropy | 800.0 | 15.9 | 34.6 | 132.7 |
| fastapi | 110.0 | 9.0 | 8.0 | 10.0 |
| accelerate | 1057.0 | 12.0 | 17.0 | 33.0 |
| transformers | 494.8 | 13.4 | 13.7 | 90.9 |
| trl | 879.0 | 5.0 | 20.0 | 352.0 |
| Liger-Kernel | 539.5 | 15.5 | 10.3 | 29.8 |
| matplotlib | 232.0 | 1.0 | 11.0 | 31.0 |
| meson | 220.0 | 7.0 | 14.0 | 16.0 |
| mlflow | 732.9 | 16.9 | 29.4 | 25.8 |
| seaborn | 518.8 | 6.4 | 25.6 | 97.1 |
| optuna | 104.0 | 12.0 | 8.0 | 28.0 |
| pandas | 1522.9 | 19.2 | 39.2 | 69.2 |
| pydantic | 436.4 | 6.8 | 24.8 | 45.6 |
| xarray | 847.8 | 15.8 | 37.3 | 21.5 |
| hatch | 847.0 | 17.0 | 55.5 | 16.0 |
| packaging | 785.0 | 3.0 | 36.0 | 294.0 |
| setuptools | 3283.0 | 50.0 | 185.0 | 30.0 |
| pytest | 528.0 | 3.0 | 34.5 | 105.5 |
| mypy | 392.0 | 11.0 | 5.0 | 10.0 |
| scikit-learn | 900.7 | 12.0 | 12.3 | 89.0 |
| sphinx | 1025.9 | 18.3 | 47.4 | 31.2 |
| sympy | 979.6 | 25.2 | 35.2 | 33.8 |

Table 12: Repository statistics.

| Repo | Summary | License |
|---|---|---|
| pytorch-lightning | Lightweight PyTorch wrapper for high-performance AI research | Apache-2.0 |
| metaflow | Framework for building and managing real-life data science projects | Apache-2.0 |
| astropy | Astronomy and astrophysics core library | BSD 3-Clause |
| fastapi | Modern, fast (high-performance) web framework for building APIs | MIT License |
| accelerate | Library for running PyTorch training on any distributed configuration | Apache-2.0 |
| transformers | State-of-the-art pretrained models for natural language processing and beyond | Apache-2.0 |
| trl | Library for training large language models with reinforcement learning from human feedback | Apache-2.0 |
| Liger-Kernel | High-performance deep learning kernels developed for large-scale distributed training | BSD 2-Clause |
| matplotlib | Plotting library for creating scientific and publication-quality visuals | Custom |
| meson | Open source build system meant to be both extremely fast and user friendly | Apache-2.0 |
| mlflow | Open source platform for the machine learning lifecycle | Apache-2.0 |
| seaborn | Statistical data visualization library built on top of matplotlib | BSD 3-Clause |
| optuna | Automatic hyperparameter optimization software framework | MIT License |
| pandas | Data analysis and manipulation library providing high-performance data structures | BSD 3-Clause |
| pydantic | Data validation using Python type hints | MIT License |
| xarray | Library for N-dimensional labeled arrays and datasets | Apache-2.0 |
| hatch | Modern, extensible Python project management | MIT License |
| packaging | Core utilities for Python packaging | Apache-2.0 |
| setuptools | Fully-featured library for packaging Python projects | MIT License |
| pytest | Testing framework for Python | MIT License |
| mypy | Optional static typing for Python | MIT License |
| scikit-learn | Machine learning algorithms and tools in Python | BSD 3-Clause |
| sphinx | Documentation generation system for Python projects | BSD 2-Clause |
| sympy | Computer algebra system for symbolic mathematics in Python | BSD 3-Clause |

Table 13: Summary and licenses for all GitHub repositories that task instances were extracted from.

| Field | Description |
|---|---|
| instance_id | (str) Unique identifier for the task. |
| patch | (str) Git diff showing the implementation. |
| test_patch | (str) Git diff showing test file modifications. |
| FAIL_TO_PASS | (list[str]) List of test files that must pass after implementation. |
| PASS_TO_PASS | (list[str]) List of test files that must continue passing. |
| image_name | (str) Docker image containing the development environment. |
| repo | (str) Source repository (e.g., "owner/repo-name"). |
| base_commit | (str) Git commit hash of the base version. |
| problem_statement | (str) Detailed task description and requirements. |
| repo_settings | (str) Repository configuration settings as JSON string. |

Table 14: Description of each field of a FeatureBench task instance object.

| Model | Repo | % Passed | % Resolved | # Token IO |
|---|---|---|---|---|
| | Liger-Kernel | 30.7 | 14.3 | 4.6M / 35k |
| | accelerate | 0.0 | 0.0 | 16.1M / 73k |
| | astropy | 34.9 | 11.8 | 12.5M / 60k |
| | fastapi | 65.0 | 0.0 | 1.0M / 14k |
| | hatch | 3.8 | 0.0 | 4.4M / 23k |
| | matplotlib | 54.0 | 0.0 | 2.0M / 40k |
| | meson | 24.4 | 0.0 | 7.0M / 57k |
| | metaflow | 100.0 | 100.0 | 4.8M / 37k |
| | mlflow | 51.4 | 28.6 | 3.0M / 20k |
| | mypy | 15.0 | 0.0 | 59k / 2k |
| Codex + | optuna | 14.3 | 0.0 | 5.1M / 36k |
| GPT-5.1-Codex | packaging | 93.5 | 0.0 | 6.0M / 45k |
| (medium reasoning) | pandas | 41.4 | 10.0 | 8.3M / 46k |
| | pydantic | 50.3 | 0.0 | 9.9M / 40k |
| | pytest | 61.8 | 0.0 | 3.2M / 38k |
| | pytorch-lightning | 51.7 | 8.3 | 8.9M / 45k |
| | scikit-learn | 84.0 | 0.0 | 7.4M / 35k |
| | seaborn | 43.7 | 8.3 | 3.3M / 41k |
| | setuptools | 23.3 | 0.0 | 4.7M / 23k |
| | sphinx | 48.4 | 20.0 | 8.1M / 49k |
| | sympy | 29.5 | 16.7 | 9.6M / 65k |
| | transformers | 24.6 | 0.0 | 4.4M / 34k |
| | trl | 38.9 | 0.0 | 1.6M / 37k |
| | xarray | 46.0 | 0.0 | 18.6M / 85k |

Table 15: Performance of Codex + GPT-5.1-Codex on each repository.

| Model | Repo | % Passed | % Resolved | # Token I/O |
|---|---|---|---|---|
| | Liger-Kernel | 77.3 | 57.1 | 3.7M / 34k |
| | accelerate | 92.3 | 0.0 | 12.2M / 53k |
| | astropy | 25.7 | 0.0 | 10.1M / 40k |
| | fastapi | 35.0 | 0.0 | 2.0M / 14k |
| | hatch | 3.8 | 0.0 | 14.7M / 54k |
| | matplotlib | 54.0 | 0.0 | 2.3M / 20k |
| | meson | 28.4 | 0.0 | 5.4M / 33k |
| | metaflow | 100.0 | 100.0 | 7.7M / 37k |
| | mlflow | 50.7 | 24.5 | 5.0M / 27k |
| | mypy | 25.0 | 0.0 | 569k / 5k |
| Claude Code | optuna | 92.9 | 0.0 | 5.8M / 37k |
| (routing) + | packaging | 88.1 | 0.0 | 9.8M / 37k |
| Claude-Opus-4.5 | pandas | 39.6 | 5.0 | 13.6M / 50k |
| | pydantic | 31.6 | 20.0 | 13.6M / 41k |
| | pytest | 67.4 | 0.0 | 8.0M / 42k |
| | pytorch-lightning | 36.2 | 0.0 | 8.8M / 35k |
| | scikit-learn | 95.2 | 33.3 | 6.6M / 35k |
| | seaborn | 42.6 | 0.0 | 5.7M / 31k |
| | setuptools | 20.0 | 0.0 | 20.1M / 73k |
| | sphinx | 39.0 | 10.0 | 10.1M / 36k |
| | sympy | 41.8 | 0.0 | 9.5M / 43k |
| | transformers | 35.6 | 2.9 | 4.4M / 26k |
| | trl | 49.9 | 0.0 | 959k / 13k |
| | xarray | 40.8 | 0.0 | 14.9M / 55k |

Table 16: Performance of Claude Code + Claude-Opus-4.5 on each repository.

| Model | Repo | % Passed | % Resolved | # Token I/O |
|---|---|---|---|---|
| | Liger-Kernel | 77.3 | 57.1 | 4.8M / 24k |
| | accelerate | 0.0 | 0.0 | 19.1M / 51k |
| | astropy | 20.2 | 0.0 | 11.1M / 35k |
| | fastapi | 80.0 | 0.0 | 3.1M / 14k |
| | hatch | 18.0 | 0.0 | 15.1M / 36k |
| | matplotlib | 65.1 | 0.0 | 2.3M / 27k |
| | meson | 29.3 | 0.0 | 4.8M / 34k |
| OpenHands + | metaflow | 100.0 | 100.0 | 5.7M / 24k |
| Claude-Opus-4.5 | mlflow | 54.1 | 20.4 | 6.9M / 26k |
| | mypy | 25.0 | 0.0 | 418k / 8k |
| | optuna | 14.3 | 0.0 | 6.2M / 23k |
| | packaging | 91.2 | 0.0 | 16.9M / 48k |
| | pandas | 50.0 | 10.0 | 10.4M / 35k |
| | pydantic | 41.9 | 0.0 | 12.2M / 30k |
| | pytest | 59.7 | 0.0 | 2.1M / 14k |
| | pytorch-lightning | 40.2 | 0.0 | 10.2M / 34k |
| | scikit-learn | 98.5 | 66.7 | 10.3M / 45k |
| | seaborn | 43.7 | 8.3 | 6.0M / 30k |
| | setuptools | 20.0 | 0.0 | 16.2M / 42k |
| | sphinx | 36.9 | 0.0 | 8.9M / 28k |
| | sympy | 50.5 | 0.0 | 11.4M / 40k |
| | transformers | 32.8 | 2.9 | 5.3M / 22k |
| | trl | 97.9 | 0.0 | 5.9M / 35k |
| | xarray | 49.7 | 0.0 | 17.1M / 47k |

Table 17: Performance of OpenHands + Claude-Opus-4.5 on each repository.

| Model | Repo | % Passed | % Resolved | # Token I/O |
|---|---|---|---|---|
| | Liger-Kernel | 63.1 | 42.9 | 2.9M / 21k |
| | accelerate | 0.0 | 0.0 | 3.4M / 23k |
| | astropy | 10.3 | 0.0 | 4.1M / 25k |
| | fastapi | 35.0 | 0.0 | 1.9M / 18k |
| | hatch | 3.8 | 0.0 | 3.4M / 23k |
| | matplotlib | 52.2 | 0.0 | 1.9M / 24k |
| | meson | 27.2 | 0.0 | 2.8M / 30k |
| OpenHands + | metaflow | 100.0 | 100.0 | 2.4M / 24k |
| DeepSeek-V3.2 | mlflow | 33.3 | 12.2 | 2.7M / 23k |
| | mypy | 25.0 | 0.0 | 532k / 7k |
| | optuna | 7.1 | 0.0 | 3.1M / 21k |
| | packaging | 32.6 | 0.0 | 4.1M / 42k |
| | pandas | 17.8 | 0.0 | 3.5M / 24k |
| | pydantic | 13.2 | 0.0 | 4.3M / 19k |
| | pytest | 12.1 | 0.0 | 4.2M / 36k |
| | pytorch-lightning | 26.2 | 0.0 | 2.9M / 24k |
| | scikit-learn | 49.8 | 0.0 | 4.8M / 26k |
| | seaborn | 27.5 | 0.0 | 2.8M / 22k |
| | setuptools | 23.3 | 0.0 | 11.4M / 51k |
| | sphinx | 29.9 | 0.0 | 2.6M / 19k |
| | sympy | 11.8 | 0.0 | 7.0M / 39k |
| | transformers | 17.9 | 2.9 | 2.3M / 19k |
| | trl | 81.4 | 0.0 | 3.0M / 54k |
| | xarray | 29.9 | 0.0 | 3.6M / 29k |

Table 18: Performance of OpenHands + DeepSeek-V3.2 on each repository.

| Model | Repo | % Passed | % Resolved | # Token I/O |
|---|---|---|---|---|
| Gemini CLI + Gemini-3-Pro-Preview (low reasoning) | Liger-Kernel | 65.9 | 42.9 | 1.0M / 15k |
| | accelerate | 0.0 | 0.0 | 2.3M / 12k |
| | astropy | 12.6 | 0.0 | 7.6M / 24k |
| | fastapi | 85.0 | 50.0 | 1.0M / 6k |
| | hatch | 3.8 | 0.0 | 1.0M / 5k |
| | matplotlib | 65.1 | 0.0 | 474k / 8k |
| | meson | 29.1 | 0.0 | 730k / 16k |
| | metaflow | 100.0 | 100.0 | 422k / 9k |
| | mlflow | 36.5 | 8.2 | 1.2M / 9k |
| | mypy | 25.0 | 0.0 | 1.0M / 3k |
| | optuna | 7.1 | 0.0 | 828k / 6k |
| | packaging | 46.6 | 0.0 | 1.2M / 11k |
| | pandas | 29.2 | 5.0 | 4.5M / 10k |
| | pydantic | 21.1 | 0.0 | 9.4M / 16k |
| | pytest | 60.0 | 0.0 | 887k / 14k |
| | pytorch-lightning | 26.4 | 0.0 | 646k / 8k |
| | scikit-learn | 81.5 | 0.0 | 1.6M / 13k |
| | seaborn | 32.9 | 0.0 | 582k / 8k |
| | setuptools | 0.0 | 0.0 | 1.4M / 8k |
| | sphinx | 35.0 | 0.0 | 2.5M / 21k |
| | sympy | 37.8 | 0.0 | 6.4M / 14k |
| | transformers | 22.3 | 0.0 | 1.4M / 9k |
| | trl | 91.2 | 0.0 | 442k / 17k |
| | xarray | 28.0 | 0.0 | 4.6M / 17k |

Table 19: Performance of Gemini CLI + Gemini-3-Pro-Preview on each repository.

| Model | Repo | % Passed | % Resolved | # Token I/O |
|---|---|---|---|---|
| OpenHands + Gemini-3-Pro-Preview (low reasoning) | Liger-Kernel | 72.6 | 42.9 | 4.5M / 41k |
| | accelerate | 0.0 | 0.0 | 4.9M / 40k |
| | astropy | 10.5 | 0.0 | 9.2M / 49k |
| | fastapi | 35.0 | 0.0 | 1.1M / 16k |
| | hatch | 3.8 | 0.0 | 7.7M / 58k |
| | matplotlib | 50.4 | 0.0 | 1.2M / 45k |
| | meson | 28.6 | 0.0 | 1.6M / 27k |
| | metaflow | 100.0 | 100.0 | 8.6M / 52k |
| | mlflow | 32.2 | 8.2 | 4.6M / 38k |
| | mypy | 25.0 | 0.0 | 479k / 10k |
| | optuna | 14.3 | 0.0 | 4.3M / 28k |
| | packaging | 0.0 | 0.0 | 12.6M / 56k |
| | pandas | 34.8 | 5.0 | 10.4M / 46k |
| | pydantic | 15.3 | 0.0 | 11.8M / 45k |
| | pytest | 67.7 | 0.0 | 16.9M / 81k |
| | pytorch-lightning | 26.5 | 0.0 | 11.1M / 36k |
| | scikit-learn | 85.9 | 0.0 | 8.3M / 48k |
| | seaborn | 30.7 | 0.0 | 2.8M / 31k |
| | setuptools | 23.3 | 0.0 | 24.3M / 71k |
| | sphinx | 30.4 | 0.0 | 5.0M / 43k |
| | sympy | 36.9 | 0.0 | 8.7M / 52k |
| | transformers | 17.0 | 0.0 | 3.5M / 31k |
| | trl | 97.3 | 0.0 | 2.1M / 58k |
| | xarray | 29.5 | 0.0 | 6.1M / 43k |

Table 20: Performance of OpenHands + Gemini-3-Pro-Preview on each repository.

| Model | Repo | % Passed | % Resolved | # Token I/O |
|---|---|---|---|---|
| | Liger-Kernel | 33.3 | 14.3 | 2.1M / 15k |
| | accelerate | 0.0 | 0.0 | 2.5M / 16k |
| | astropy | 5.7 | 0.0 | 2.4M / 13k |
| | fastapi | 35.0 | 0.0 | 499k / 5k |
| | hatch | 3.8 | 0.0 | 1.3M / 12k |
| | matplotlib | 43.1 | 0.0 | 888k / 12k |
| | meson | 28.6 | 0.0 | 1.5M / 15k |
| OpenHands + | metaflow | 100.0 | 100.0 | 1.2M / 8k |
| Qwen3-Coder-480B- | mlflow | 36.2 | 10.2 | 1.9M / 14k |
| A35B-Instruct | mypy | 25.0 | 0.0 | 284k / 3k |
| | optuna | 7.1 | 0.0 | 951k / 8k |
| | packaging | 16.7 | 0.0 | 5.3M / 39k |
| | pandas | 20.2 | 0.0 | 2.2M / 13k |
| | pydantic | 13.1 | 0.0 | 2.7M / 16k |
| | pytest | 52.9 | 0.0 | 3.2M / 19k |
| | pytorch-lightning | 26.1 | 0.0 | 1.4M / 13k |
| | scikit-learn | 83.1 | 0.0 | 2.9M / 24k |
| | seaborn | 27.6 | 0.0 | 2.0M / 15k |
| | setuptools | 10.0 | 0.0 | 1.4M / 8k |
| | sphinx | 28.9 | 0.0 | 1.7M / 11k |
| | sympy | 12.6 | 0.0 | 4.5M / 27k |
| | transformers | 8.8 | 0.0 | 1.7M / 12k |
| | trl | 70.3 | 0.0 | 1.6M / 40k |
| | xarray | 21.0 | 0.0 | 1.8M / 21k |

Table 21: Performance of OpenHands + Qwen3-Coder-480B-A35B-Instruct on each repository.

**Output Analysis of Gemini 3 Pro**

```
{
  "timestamp": ...,
  "model": "gemini-3-pro-preview",
  "tools": [
    {
      "functionDeclarations": [
        {
          "name": "read_file",
          "description": "Reads and returns the content of a
              specified file...",
          "parametersJsonSchema": {
            "type": "object",
            "required": [
              "file_path"
            ],
            "properties": {
              "file_path": {
                "description": "The path to the file to read.",
                "type": "string"
              },
              "limit": ...
              "offset": ...
            ...],
  "request": {
    "contents": [
      ...,
      {
        "role": "model",
        "parts": [
          {
            "thoughtSignature": "...",
            "functionCall": {
              "name": "read_file",
              "args": {
                "path":"/testbed/astropy/visualization/basic_rgb.py
                        ",
              ...},
      {
        "role": "user",
        "parts": [
          {
            "functionResponse": {
              "name": "read_file",
              "response": {
                "error": "params must have required property '
                    file_path'"
              ...},
  "response": {
    "status": 200,
    "content": "**Clarifying File Requirement**\n\nI've realized my
        initial approach was incomplete. Specifically, I understand
        now that the 'read_file' tool explicitly needs a 'file_path'
         input. I'm focusing on ensuring that parameter is
        accurately and appropriately utilized moving forward."
  }
}
```

Figure 8: Partial output of Gemini 3 Pro model using Gemini CLI framework while completing task
`astropy__astropy.b0db0daa.test_basic_rgb.067e927c.lv1`

**Prompt for classifying top-level objects part 1 of 2**

```
Task: From a list of candidate objects, identify which ones are "
    tested objects" in the context of a Python test file.

**Definition of "Tested Object":**
A "tested object" is an object that the test file is specifically
    designed to test. It represents the core functionality or
    feature being validated, NOT utility functions, test helpers, or
     infrastructure code.

**Test File Information:**
- Test file path: `{test_file}`
- Test file name: `{test_file_name}`

**Test File Content:**
```python
{test_file_content}
```

**Candidate Objects to Classify:**

{candidates_section}

**Classification Guidelines:**

**Tested Objects (should be selected):**
- Core algorithms, classes, or functions that the test file is
    designed to validate
- Main interfaces or APIs being tested
- Key components whose behavior is the primary focus of the test

**Non-Tested Objects (should NOT be selected):**
- Utility functions from test utilities (e.g., `test.utils.*`, `
    pytest.*`)
- Common tools defined in the codebase (e.g., `infer_device()`, `
    assert_verbose_allclose()`)
```

Figure 9: Prompt template for classifying top-level objects part 1 of 2

**Prompt for classifying top-level objects part 2 of 2**

```
**Examples for Reference:**

**Example Scenario:**
- Test file: '/testbed/test/transformers/test_jsd.py' (testing JSD
    algorithm)

**Should Select (Tested Objects):**
- '/testbed/src/liger_kernel/transformers/jsd.py::LigerJSD.forward
    ::64' - Core JSD implementation
- '/testbed/src/liger_kernel/transformers/jsd.py::LigerJSD.__init__
    ::59' - JSD class initialization

**Should NOT Select (Non-Tested Objects):**
- '/testbed/test/transformers/test_jsd.py::_test_correctness_once
    ::91' - Helper function in test file
- '/testbed/src/liger_kernel/ops/utils.py::ensure_contiguous.wrapper
    ::34' - General utility

**Your Task:**
Please analyze each candidate object and determine which ones are
    tested objects for the given test file.
Provide your response in the following structured format:

## Analysis
For each candidate object, briefly explain whether it should be
    selected as a tested object and why.

## Final Answer
Provide your final selection in the following JSON format:
'''json
{{
    "tested_object_ids": [
        "object_id_1",
        "object_id_2"
    ],
    "reasoning": "Brief summary of the selection criteria applied"
}}
'''
**Important Notes:**
- The 'tested_object_ids' list should contain ONLY the object IDs
    that are tested objects
- If none of the candidates are tested objects, return an empty list
    : '"tested_object_ids": []'
- Include the full object ID exactly as provided in the candidate
    list
- Unless it's obvious or you're pretty sure that a candidate object
    is a general purpose tool, you need to categorize it as tested
    object

Now, please begin your analysis for the candidate objects listed
    above.
```

Figure 10: Prompt template for classifying top-level objects part 2 of 2

**Prompt for completing docstring**

```
Generate a detailed docstring for the following Python function. The
    docstring should include:
1. The main function description
2. Parameter description (if any)
3. Return value description (if any)
4. Important notes or exceptions (if applicable)

Function qualified name:
```python
{function_qualified_name}
```

The full file content for reference:
```python
{file_content}
```

You only need to generate the docstring of the {
    function_qualified_name} function, the complete content of the
    file is for your reference only, and there is no need to
    generate the docstring of other functions or classes.

In docstring, in order for us to parse it correctly, you are
    forbidden to use syntax like ```python ```, which may cause the
    end result to be confusing. Please only return the docstring
    content!!! DO NOT include triple quotes or other format tags:
```

Figure 11: Prompt template for completing docstring given to LLM

**User prompt part 1 of 3 (Level 1)**

```
## Task
**Task Statement:**

Implement ...

1. **Core functionalities**: ...

2. **Main features and requirements**:
   - ...
   - ...

3. **Key challenges**:
   - ...
   - ...
```

Figure 12: Unified prompt template for $L_1$ part 1, Task.

---

**User prompt part 2 of 3 (Level 1)**

```
**NOTE**:
- This test comes from the `{{ library_name }}` library, and we have
     given you the content of this code repository under `/testbed
     /`, and you need to complete based on this code repository and
     supplement the files we specify. Remember, all your changes must
      be in this codebase, and changes that are not in this codebase
     will not be discovered and tested by us.
- We've already installed all the environments and dependencies you
     need, you don't need to install any dependencies, just focus on
     writing the code!
- **CRITICAL REQUIREMENT**: After completing the task, pytest will
     be used to test your implementation. **YOU MUST** match the
     exact interface shown in the **Interface Description** (I will
     give you this later)

You are forbidden to access the following URLs:
{{ black_links }}

Your final deliverable should be code under the `/testbed/`
     directory, and after completing the codebase, we will evaluate
     your completion and it is important that you complete our tasks
     with integrity and precision.

The final structure is like below.

 /testbed     # all your work should be put into this codebase
   ├── dir1/
   │    ├── file1.py
   │    ├── ...
   ├── dir2/

## Interface Descriptions

### Clarification
The **Interface Description**  describes what the functions we are
    testing do and the input and output formats.

for example, you will get things like this:

Path: `{{ interface_code_example_path }}`
```python
{{ interface_code_example }}
```

The value of Path declares the path under which the following
    interface should be implemented and you must generate the
    interface class/function given to you under the specified path.
```

Figure 13: Unified prompt template for $L_1$ part 2, Precautions.

**User prompt part 3 of 3**

In addition to the above path requirement, you may try to modify any
    file in codebase that you feel will help you accomplish our
    task. However, please note that you may cause our test to fail
    if you arbitrarily modify or delete some generic functions in
    existing files, so please be careful in completing your work.

What's more, in order to implement this functionality, some
    additional libraries etc. are often required, I don't restrict
    you to any libraries, you need to think about what dependencies
    you might need and fetch and install and call them yourself. The
     only thing is that you **MUST** fulfill the input/output format
     described by this interface, otherwise the test will not pass
    and you will get zero points for this feature.

And note that there may be not only one **Interface Description**,
    you should match all **Interface Description {n}**

### Interface Description 1
Below is **Interface Description 1**

Path: '/path/to/xxx.py'
```python
def my_function:
    """
    Implement a ...

    Parameters:
    arg1: str, ...

    Returens:
    arg2: int, ...
    ...
    """
    # <your code>
```

### Interface Description 2
class MyClass:
    """

    ...
    """
    # <your code>
...

Remember, **the interface template above is extremely important**.
    You must generate callable interfaces strictly according to the
    specified requirements, as this will directly determine whether
    you can pass our tests. If your implementation has incorrect
    naming or improper input/output formats, it may directly result
    in a 0% pass rate for this case.

Figure 14: Unified prompt template for $L_1$ part 3, Test and Interface Description.

---

**User prompt for test-layer-norm (Level 1)**

```
## Task
**Task Statement: Implement Optimized Layer Normalization Function**

Develop a high-performance layer normalization function that:

1. **Core Functionality**: Applies layer normalization to input
    tensors using weight, bias, and epsilon parameters for numerical
     stability

2. **Key Requirements**:
   - Integrate with existing Liger kernel optimization framework
   - Support standard layer normalization mathematical operations (
       mean centering, variance scaling, affine transformation)
   - Handle multi-dimensional tensor inputs efficiently

3. **Main Challenges**:
   - Optimize memory usage and computational performance
   - Ensure numerical stability with configurable epsilon values
   - Maintain compatibility with transformer model architectures
   - Provide seamless integration with the broader Liger functional
       interface ecosystem

**NOTE**:
- This test comes from the 'liger-kernel' library, and we have given
     you the content of this code repository under '/testbed/', and
    you need to complete based on this code repository and
    supplement the files we specify. Remember, all your changes must
     be in this codebase, and changes that are not in this codebase
    will not be discovered and tested by us.
- We've already installed all the environments and dependencies you
    need, you don't need to install any dependencies, just focus on
    writing the code!
- **CRITICAL REQUIREMENT**: After completing the task, pytest will
    be used to test your implementation. **YOU MUST** match the
    exact interface shown in the **Interface Description** (I will
    give you this later)

You are forbidden to access the following URLs:
black_links:
- https://github.com/linkedin/Liger-Kernel/

Your final deliverable should be code under the '/testbed/'
    directory, and after completing the codebase, we will evaluate
    your completion and it is important that you complete our tasks
    with integrity and precision.

The final structure is like below.

 /testbed      # all your work should be put into this codebase
  ├── dir1/
  │    ├── file1.py
  │    ├── ...
  ├── dir2/

## Interface Descriptions
...
```

Figure 15: User prompt for test-layer-norm ($L_1$).

---

**User prompt for test-layer-norm (Level 2) part 1 of 2**

```
## Task
**Task Statement: Implement Optimized Layer Normalization Function**

Develop a high-performance layer normalization function that:

1. **Core Functionality**: Applies layer normalization to input
    tensors using weight, bias, and epsilon parameters for numerical
     stability

2. **Key Requirements**:
   - Integrate with existing Liger kernel optimization framework
   - Support standard layer normalization mathematical operations (
       mean centering, variance scaling, affine transformation)
   - Handle multi-dimensional tensor inputs efficiently

3. **Main Challenges**:
   - Optimize memory usage and computational performance
   - Ensure numerical stability with configurable epsilon values
   - Maintain compatibility with transformer model architectures
   - Provide seamless integration with the broader Liger functional
       interface ecosystem

**NOTE**:
- This test is derived from the 'liger-kernel' library, but you are
    NOT allowed to view this codebase or call any of its interfaces.
     It is **VERY IMPORTANT** to note that if we detect any viewing
    or calling of this codebase, you will receive a ZERO for this
    review.
- **CRITICAL**: This task is derived from 'liger-kernel', but you **
    MUST** implement the task description independently. It is **
    ABSOLUTELY FORBIDDEN** to use 'pip install liger-kernel' or some
     similar commands to access the original implementation, and
    doing so will be considered cheating and will result in an
    immediate score of ZERO! You must keep this firmly in mind
    throughout your implementation.
- You are now in '/testbed/', and originally there was a specific
    implementation of 'liger-kernel' under '/testbed/' that had been
     installed via 'pip install -e .'. However, to prevent you from
    cheating, we've removed the code under '/testbed/'. While you
    can see traces of the installation via the pip show, it's an
    artifact, and 'liger-kernel' doesn't exist. So you can't and don
    't need to use 'pip install liger-kernel', just focus on writing
     your 'agent_code' and accomplishing our task.
- Also, don't try to 'pip uninstall liger-kernel' even if the actual
     'liger-kernel' has already been deleted by us, as this will
    affect our evaluation of you, and uninstalling the residual '
    liger-kernel' will result in you getting a ZERO because our
    tests won't run.
- We've already installed all the environments and dependencies you
    need, you don't need to install any dependencies, just focus on
    writing the code!
- **CRITICAL REQUIREMENT**: After completing the task, pytest will
    be used to test your implementation. **YOU MUST** match the
    exact interface shown in the **Interface Description** (I will
    give you this later)
```

Figure 16: User prompt for test-layer-norm part 1 of 2 ($L_2$).

**User prompt for test-layer-norm (Level 2) part 2 of 2**

```
You are forbidden to access the following URLs:
black_links:
- https://github.com/linkedin/Liger-Kernel/

Your final deliverable should be code in the `/testbed/agent_code`
    directory.
The final structure is like below, note that all dirs and files
    under agent_code/ are just examples, you will need to organize
    your own reasonable project structure to complete our tasks.
```

```
 /testbed
  ├── agent_code/        # all your code should be put into this dir and match the specific dir structure
  │   ├── __init__.py    # `agent_code/` folder must contain `__init__.py`
  │   ├── dir1/
  │   │   ├── __init__.py
  │   │   ├── code1.py
  │   │   ├── …
  ├── setup.py           # after finishing your work, you MUST generate this file
```

```
After you have done all your work, you need to complete three
    CRITICAL things:
1. You need to generate `__init__.py` under the `agent_code/` folder
    and import all the classes or functions described in the **
    Interface Descriptions** in it. The purpose of this is that we
    will be able to access the interface code you wrote directly
    through `agent_code.ExampleClass()` in this way.
2. You need to generate `/testbed/setup.py` under `/testbed/` and
    place the following content exactly:
```python
from setuptools import setup, find_packages
setup(
    name="agent_code",
    version="0.1",
    packages=find_packages(),
)
```
3. After you have done above two things, you need to use `cd /
    testbed && pip install .` command to install your code.
Remember, these things are **VERY IMPORTANT**, as they will directly
     affect whether you can pass our tests.

## Interface Descriptions
...
```

Figure 17: User prompt for test-layer-norm part 2 of 2 ($L_2$).

**Example of an LLM-generated docstring**

```
class LigerFusedLinearDPOFunction(LigerFusedLinearPreferenceBase):
    ...
    @staticmethod
    def backward(ctx, *grad_output):
        """
        Computes gradients for the fused linear DPO (Direct
            Preference Optimization) function during backpropagation
            .

        This static method implements the backward pass for
            automatic differentiation in PyTorch's autograd system.
            It delegates the gradient computation to the parent
            class's backward method and then filters the results to
            match the expected number of parameters for the DPO
            function.

        Parameters:
            ctx: PyTorch autograd context object containing saved
                tensors and other information from the forward pass
            *grad_output: Variable-length argument list of gradient
                tensors flowing back from the loss function. Each
                tensor represents the gradient with respect to the
                corresponding output of the forward pass

        Returns:
            tuple: A tuple containing gradients with respect to the
                input parameters of the forward function:
                - Gradient w.r.t. input tensor (_input)
                - Gradient w.r.t. weight tensor (weight)
                - Gradient w.r.t. target tensor (target)
                - Gradient w.r.t. bias tensor (bias)
                - None values for parameters that don't require
                    gradients (ref_input, ref_weight, ref_bias,
                    ignore_index, beta, compute_nll_loss, compiled,
                    use_ref_model, average_log_prob, chunk_size,
                    loss_type)

        Important Notes:
            - This method is part of PyTorch's Function interface
                for custom autograd operations
            - The method truncates the parent class gradients to the
                first 4 elements using [:4] slicing
            - Additional None values are returned to match the
                signature of the forward method parameters
            - The actual gradient computation logic is inherited
                from LigerFusedLinearPreferenceBase.backward()
            - This ensures proper gradient flow for DPO loss
                optimization while maintaining compatibility with
                PyTorch's autograd system
        """
        <your code>
    ...
```

Figure 18: Example of an LLM-generated docstring.

---

**Human Evaluation Guideline for Top-Level Tested Object Classification**

Each import statement in a test file should be evaluated independently to determine whether it represents a **top-level tested object** or an **auxiliary component**. The procedure is as follows:

**Step 1: Understand Test File Purpose**

- Read the test file to understand its testing objective and scope.
- Identify the main functionality or module being validated.

**Step 2: Identify All Import Statements**

- Locate all import statements, including absolute and relative imports.

**Step 3: Filter External Library Imports**

- Exclude imports from external libraries (e.g., `pytest`, `unittest`, `torch`).

**Step 4: Classify Repository-Internal Imports**

- **Assert statement usage:** If the imported object appears in assertions comparing results, it is likely a tested object.
- **Name correspondence:** If the object's name matches keywords in the test filename, it is likely a tested object.
- **Module correspondence:** If imported from a module matching the test filename, it is likely a tested object.
- **Utility module exclusion:** Imports from `utils/`, `testing/`, `helpers/`, etc., are usually auxiliary.
- **Frequency and prominence:** Objects used extensively across the test file are more likely tested objects.

**Classification Decision**

Mark each import as either a **Top Import (tested object)** or **Non-Top Import** (auxiliary). When criteria conflict, prioritize the first three criteria over the last two.

---

Figure 19: Human evaluation guideline for identifying top-level tested objects.

---

**Expert Verification Guideline for Feature-Level Tasks (part 1 of 2)**

Each feature-level task must be manually verified to ensure that (1) the task is structurally correct (objects, imports, masking, etc.), and (2) a competent engineer can implement the required functionality using *only* the prompt and the remaining codebase, without external documentation.

You will typically use two resources:

- `metadata_outputs/`: logs and classification results (lists of top objects and specific objects).
- Level-1 task directory: `problem_statement.md` (this is the primary target for verification).

**Stage 1: Check Structural Consistency**
**1.1 Get the list of top and specific objects**

- Open the classification summary under `metadata_outputs/`.
- Identify:
  - *Top objects*: top interface of the feature being tested.
  - *Specific objects*: objects that are functionally related but are not top interfaces.
- Treat these lists as a checklist for the following steps.

**1.2 Check masking of top objects**

- For each top object, open its source file in the Level-1 directory.
- Confirm that the implementation body is removed and that only the signature, docstring, and minimal scaffolding remain.
- If any implementation detail is still visible, manually remove it while ensuring that tests can still import and call the interface.

**1.3 Check removing of specific objects**

- For each specific object, confirm that it does not appear in the remain codebase.
- If leftover definitions are found, please remove them.

---

Figure 20: Expert verification guideline for feature-level tasks part 1.

---

**Expert Verification Guideline for Feature-Level Tasks (part 2 of 2)**

**Stage 2: Check Prompt Completeness and Solvability**
**2.1 Check the high-level Task Description**

- Read the Task Description and ask: *"If I only had this description and the codebase, do I know what to implement?"*
- Verify that it:
    - Explains what feature or behavior needs to be implemented.
    - Provides context about where the feature sits in the system.
    - Mentions any key technical considerations that affect correctness.
- If the description is vague or incomplete, rewrite it to make the implementation goal clear.

**2.2 Check the Test Description sections**

- For each Test Description, verify that:
    - All required top objects are correctly referenced.
    - Imports match the real file structure under the task directory.
- Fix missing interfaces and incorrect module paths as needed so that the agent will know where to implemenet them.

**2.3 Check Interface Descriptions and docstrings**

- For each Interface Description:
    - Ensure the docstring is semantically complete: what the function/class does, parameter meanings, and return values.
    - Confirm that it is self-contained and does not require external documentation.
    - Keep it concise but readable; something a real engineer would be happy to follow.
- If a docstring is too short, ambiguous, or inconsistent with the real behavior, revise it and, if necessary, refer to the original implementation to understand the intended semantics.

**When to Mark a Task as Verified**
A task is considered **verified** if a competent engineer can implement the required functionality using only the prompt.md and the remained codebase without external documentation. Concretely, this requires that:

- All top objects are correctly masked, imported, and documented.
- All specific objects that should not remain in the codebase are removed.
- The Task Description clearly states what to build.
- Test Descriptions match the actual file layout and cover all required interfaces.
- Interface Descriptions and docstrings are accurate and self-contained.

If any of these conditions are not met, fix the relevant parts and re-check the task using the same steps before marking it as verified.

Figure 21: Expert verification guideline for feature-level tasks part 2.

