# OpenReview forum: "FeatureBench: Benchmarking Agentic Coding for Complex Feature Development"
_ICLR.cc/2026/Conference — ICLR 2026 Poster_

### Official Review · Reviewer_3aZ5 · 2025-10-25

**Soundness:** 3
**Presentation:** 3
**Contribution:** 3
**Rating:** 6
**Confidence:** 4

**Summary:**

This paper introduces ACE-Bench, a new benchmark for evaluating LLM-based agents on feature-oriented software development tasks. Unlike existing benchmarks focusing primarily on bug-fixing within single PRs, ACE-Bench proposes more complete coding scenarios. The benchmark comprises 212 tasks and 889 executable environments from 16 open-source Python repositories.

Key contributions include:

(1) A feature-oriented evaluation framework with two difficulty levels (L1: extending existing codebases; L2: implementing from scratch)

(2) Execution-based evaluation with explicit interface specifications to enable unambiguous testing

(3) Empirical results show that more capable models achieve very low success rate, like Claude 4 Sonnet achieves only 7.5% success rate

**Strengths:**

**1. Problem Formulation**: This work proposes ACE-Bench that contains complex software engineering tasks. Its problem formulation explains its differences from existing benchmarks in task complexity and dataset construction.

**2. Rigorous Evaluation Protocol**: The execution-based evaluation with comprehensive anti-cheating mechanisms and two difficulty levels provides a reliable performance assessment.

**3. Significant Challenge**: The obvious performance drop between SWE-bench and ACE-Bench demonstrates the limitations of current agentic systems, providing meaningful direction for future research.

**Weaknesses:**

**1. Limitations In Scale and Diversity**:

- ACE-Bench uses 212 evaluation data from 16 repositories, which can be relatively small.
- Python-only instances limits generalizability to other programming languages and real-world scenarios.
- Repository selection criteria are not clearly justified.

**2. Limitations In Methodology**:

- Using LLMs to classify top-level objects introduces potential systematic biases and errors. However, no quantitative evaluation of classification accuracy is provided.
- The 100-line minimum and "10 F2P test points" filtering criteria lack justification.
- The paper sets m=5 P2P tests per instance, but doesn't justify why this number provides adequate coverage.

**3. Limitations In Evaluation**:

- The average of 1M+ input tokens per task raises concerns about practical applicability and cost.
- Near-zero success rates on L2 tasks suggest the difficulty may be unrealistically high, limiting the benchmark's ability to differentiate between models and provide meaningful insights for future research.
- Figure 5 shows minimal correlation between task creation time and performance, but deeper analysis of how feature complexity evolves over time can be valuable.

**4. Limitations In Interface Specification Dependency**:

- Table 7 shows remarkable performance drops without interfaces, suggesting that the benchmark may be testing interface-matching more than general coding ability.
- Real-world development often involves ambiguous or evolving requirements, while the explicit interface specification may not reflect realistic scenarios.

**Questions:**

My questions are following several aspects mentioned in weakness:

- How can you ensure the generalizability of ACE-Bench, given various programming languages and real-world scenarios?
- What are your repository selection criteria, and why do you believe 16 repositories are sufficient?
- What are the justifications and verifications for LLM classification, filtering criteria, and the number setting to ensure adequate coverage?
- How would you address the limitations in evaluation, such as the high input tokens and cost, undistinguishable performance, and inadequate analysis?
- How are you going to address these interface specification dependency problems in ACE-Bench?

---

> ### Author Response · Authors · 2025-11-21
> **Response to Reviewer 3aZ5 (1/3)**
>
> We sincerely appreciate your recognition of the importance of our work and thank you for the thoughtful and constructive feedback. Our responses are provided below.
>
> ---
>
> **Q1:** **ACE-Bench uses 212 evaluation data from 16 repositories, which can be relatively small. and why do you believe 16 repositories are sufficient?**
>
> **A1:** The initial benchmark size was chosen with evaluation cost in mind. As noted in later questions, running experiments on ACE-Bench is computationally expensive, and a substantially larger dataset could impose a heavy burden on the community, reducing its practical usability. Nevertheless, our pipeline scales efficiently. For example, in our internal test, we expanded the dataset by selecting 18 additional repositories and generated 634 new cases within two days. To balance accessibility and comprehensiveness, we plan to release three benchmark tiers: a lite version with 30 instances, a medium version with 212 instances (the current full release), and a large version with 1,000 instances.
>
> ---
>
> **Q2:** **Python-only instances limits generalizability to other programming languages and real-world scenarios. How can you ensure the generalizability of ACE-Bench, given various programming languages and real-world scenarios?**
>
> **A2:** We appreciate this insightful comment. Our current focus on Python stems from its dominant role in open-source software and the availability of well-maintained repositories with strong test coverage. This aligns ACE-Bench with prior repository-level benchmarks such as SWE-Bench, ensuring comparability and accessibility. Python’s popularity among AI researchers also simplifies human verification and error analysis.
>
> Prior work demonstrates that model performance on Python benchmarks is highly correlated with that in other languages [1,2], supporting Python as a reliable proxy for general coding ability.
>
> Our pipeline is designed to be language-agnostic and can be extended to other languages with minimal engineering effort. For example, extending to Java would only require replacing pytest based test discovery with JUnit discovery, and replacing Python tracing with a small Java agent attached during mvn test.
>
> [1] Rozière, Baptiste, et al. "Code Llama: Open Foundation Models for Code.", 2023, http://arxiv.org/abs/2308.12950. arXiv.
>
> [2] Cassano, Federico, et al. "MultiPL-E: A Scalable and Extensible Approach to Benchmarking Neural Code Generation", 2022, http://arxiv.org/abs/2208.08227. arXiv.
>
> ---
>
> **Q3:** **Repository** **selection criteria are not clearly justified. What are your repository selection criteria?**
>
> **A3:** We aligned our repository selection with SWE-Bench's choices, for SWE-Bench is an outstanding work widely recognized by the community. Meanwhile, considering the widespread use of AI at present, we have selected some highly-starred repositories from the AI domain beyond the SWE-Bench's choices. In subsequent data releases, we will consider more comprehensive criteria to ensure the representativeness and quality of the selected repositories—this will be achievable thanks to the scalability of our data pipeline.
>
> ---
>
> **Q4:** **Using** **LLMs** **to classify top-level objects introduces potential systematic biases and errors. However, no quantitative evaluation of classification accuracy is provided.**
>
> **A4:** We experimented with several rule-based approaches for identifying top-level objects in test files, including string matching and structural heuristics, but none generalized reliably across repositories. Modern LLMs, however, handle this classification task very well in practice, which is why we adopted an LLM-based approach. We agree that quantitative evidence is necessary. So we conducted a quantitative evaluation against expert annotations. LLM achieved 91.74% accuracy out of 605 import statements in the lite set. For more categorized metrics, please check the table below. Detailed analysis can be found in Section 4.2.1 of the revised manuscript.
>
> | Metrics  | Accuracy | Recall | precision | F1 Score |
>   | -------- | -------- | ------ | --------- | -------- |
>   | Value(%) | 91.7     | 89.2   | 81.3      | 84.9     |

---

> ### Author Response · Authors · 2025-11-21
> **Response to Reviewer 3aZ5 (2/3)**
>
> **Q5:** **The 100-line minimum and "10 F2P test points" filtering criteria lack justification.**
>
> **A5:** Our filtering thresholds follow common scales observed in existing benchmarks. Prior work shows that real feature development typically involves substantial code changes: SWE-Bench Pro [1] reports an average of 107.4 modified lines, and DevEval [2] reports 112 lines. We therefore require at least 100 lines of relevant code to ensure that each task reflects meaningful feature-level complexity.
>
> We set a minimum of 10 F2P test points to ensure that the failing cases capture enough aspects of the feature to make the task meaningful. In SWE-Bench, the average number of F2P test points is 9.1, which supports our choice of 10 as a reasonable lower bound.
>
> [1] Deng, Xiang, et al. "SWE-Bench Pro: Can AI Agents Solve Long-Horizon Software Engineering Tasks?.", 2025, https://arxiv.org/pdf/2509.16941, arXiv.
>
> [2] Li, Bowen, et al. "Prompting large language models to tackle the full software development lifecycle: A case study.", 2024, ACL.
>
> ---
>
> **Q6:** **The paper sets m=5** **P2P** **tests per instance, but doesn't justify why this number provides adequate coverage.**
>
> **A6:** We first clarify that in ACE-Bench, F2P and P2P refer to test files, each of which may contain many individual test points. The wording in Table 3 in our main paper may have caused confusion, and we have corrected it in the revision.
>
> Selecting five P2P test files balances coverage with the cost of data generation and evaluation. Running the entire test suite for every instance would make both construction and evaluation prohibitively expensive. In SWE-Bench, the average number of P2P test points is 111.7. In ACE-Bench, selecting five P2P files yields an average of 314 preserved test points, which provides strong behavioral coverage while keeping the pipeline practical.
>
> ---
>
> **Q7:** **The average of 1M+ input tokens per task raises concerns about practical applicability and cost. How would you address the high input tokens and cost?**
>
> **A7:** We agree that evaluation cost is a real concern for project scale coding benchmarks. This issue also appears in prior work, such as PaperBench [1], where a single instance exceeds 50M input tokens. To improve practicality, we provide a lite subset of 30 tasks. As shown in the table below,  performance on the lite subset closely matches that on the full benchmark. This makes the lite set a practical and lower cost option that still offers reliable performance indicators.
>
> | Model                          | % Passed (Lite) | % Resolved (Lite) | % Passed (Full) | % Resolved (Full) |
> | ------------------------------ | --------------- | ----------------- | --------------- | ----------------- |
> | Gemini 2.5 Pro                 | 17.0            | 3.3               | 13.2            | 2.4               |
> | OpenAI o3                      | 23.2            | 3.3               | 22.4            | 1.9               |
> | Qwen3-Coder-480B-A35B-Instruct | 25.6            | 0.0               | 25.4            | 2.4               |
> | GPT-5                          | 29.5            | 6.7               | 36.4            | 7.1               |
> | Claude Sonnet 4                | 37.0            | 6.7               | 38.2            | 7.5               |
>
> [1] Starace, Giulio, et al. "PaperBench: Evaluating AI's Ability to Replicate AI Research.", 2025, https://arxiv.org/pdf/2504.01848, arXiv.
>
> ---
>
> **Q8:** **Near-zero success rates on L2 tasks suggest the difficulty may be unrealistically high, limiting the benchmark's ability to differentiate between models and provide meaningful** **insights** **for future research. How would you address the undistinguishable performance?**
>
> **A8:** The very low success rate on L2 tasks is expected. L2 requires implementing the feature entirely from scratch, which is substantially harder than L1. L1 already provides a clear separation between models. For L2, we also report the pass rate, which offers a more fine-grained and informative signal even when overall success rates are low.
>
> We acknowledge that L2 is extremely challenging for current models. It reflects a real scenario where no prior code is available. Similar patterns have appeared before: early results on SWE-Bench Lite were also in the single digits (4.33%), yet models now exceed sixty percent (60.33%). We expect similar progress over time. Maintaining a challenging from-scratch setting provides long-term headroom for research and helps guide the development of more capable coding agents.

---

> ### Author Response · Authors · 2025-11-21
> **Response to Reviewer 3aZ5 (3/3)**
>
> **Q9:** **Figure 5 shows minimal correlation between task creation time and performance, but deeper analysis of how feature complexity evolves over time can be valuable.**
>
> **A9:** Thank you for the suggestion. We agree that examining how feature complexity evolves over time is valuable. In the revised manuscript Section 4.2.1, we include a more detailed analysis and plot the trend of code complexity over commit time. The trend figure shows no apparent relationship between commit time and functional complexity, while the variation in task performance is driven far more by feature complexity than by commit time.
>
> ---
>
> **Q10**: **Table 7 shows remarkable performance drops without interfaces, suggesting that the benchmark may be testing interface-matching more than general coding ability.**
>
> **A10**: The performance drop in Table 7 is expected because matching the interface is a prerequisite for passing unit tests. Without the interface specification, an agent may implement the correct functionality but still fail due to mismatched entry points. And it should be clarified that we have only given the interface to the top object (i.e., the object that is directly imported by the test file), and there is a large amount of core code which we have not given the interface needs to to be implemented by agent. Moreover, matching a prescribed interface is itself a core component of general coding ability.
>
> At the same time, the evaluation results also show that the benchmark is not testing interface-matching only. Even with the full interface, call path, and annotations provided, the strongest current agent achieves only a 7.5% resolve rate. To further validate this, we manually inspected all failing test points in the lite subset and found that only 12.54% of failures were due to interface mismatches, with most of these interface issues falling into `Missing attribute` and `TypeError` cases. Most failures instead came from errors in the underlying feature logic. This indicates that solving an ACE-Bench task requires not only aligning with the interface but also correctly implementing the substantial amount of underlying feature logic.
>
> ---
>
> **Q11**: **Real-world development often involves ambiguous or evolving requirements, while the explicit interface specification may not reflect realistic scenarios.**
>
> **A11**: We agree that real development often involves ambiguous or evolving requirements. At the same time, interface-driven development is also a common and important pattern in large engineering projects. In many industrial workflows, a feature does not begin with an entirely vague natural language description. Instead, teams typically discuss the requirement, then finalize concrete interface specifications, expected entry points, and minimal behavioral notes before implementation. Developers then write the internal logic under these agreed interfaces.
>
> We recognize that developing features directly from vague natural language is also valuable and realistic. This requires addressing the gap between high-level descriptions, concrete interface design, and full implementations. Studying this direction is meaningful, and we plan to extend ACE-Bench in future versions to study this capability.
>
> In addition, the information in our current prompt is complete, and if we want to get ambiguous information, we only need to destroy our prompt, or remove the interface description part in the prompt. For evolving requirements, this scenario can be simulated by first giving the corrupted prompt and then giving the complete prompt after the agent completes the task. Thank you for your valuable insights, and we may conduct deeper experimental explorations on these points in the future.

---

> > ### Comment · Reviewer_3aZ5 · 2025-11-27
> >
> > Thank you for your detailed response! Also look forward to future improvements in generalizability, such as broader coverage of different programming languages and real-world scenarios. The `Missing attribute` and `TypeError` cases that you have mentioned are interesting patterns, and could benefit from deeper discussion and a more comprehensive classification of error types.

---

> > > ### Author Response · Authors · 2025-11-28
> > > **Response to Reviewer 3az5 (1/2)**
> > >
> > > Thank you for your constructive feedback and for highlighting generalizability as an important direction for future work. We plan to extend ACE-Bench to cover more programming languages and more realistic development scenarios, including high-level natural-language feature requests, in future releases.
> > >
> > > Regarding error-type classification, Figure 4 in our paper presents the error distribution of *Claude 4 Sonnet* on the full dataset, and we include the corresponding table below for clarity. Based on this table and our manual inspection of agent execution trajectories, we identify two primary failure modes:
> > >
> > > | Failure Modes | TypeError | AssertionError | ValueError | RuntimeError | AttributeError | Others |
> > > | ------------- | --------- | -------------- | ---------- | ------------ | -------------- | ------ |
> > > | % Proportion  | **29.70** | **29.59**      | 12.54      | 10.88        | 9.22           | 8.07   |
> > >
> > > Based on the table and our manual inspection of agent execution trajectories, we identify two main failure modes:
> > >
> > > - `TypeError` accounts for the largest proportion of failures, indicating that current LLMs often tend to guess (even hallucinate) the interface of a function defined across files, rather than performing the related file access to retrieve the precise function signatures.
> > > - `AssertionError` represents the second most common category, indicating that many generated solutions can run successfully up to the assertion checkpoints without triggering earlier runtime errors such as TypeError. This pattern highlights a core limitation of current LLMs: while they can produce implementations that conform to the specified interfaces, they frequently fail to capture the correct underlying semantic logic.
> > >
> > > Overall, these findings are consistent with our expectations: while models can follow the given interfaces, their limitations in semantic reasoning and in understanding the argument structures of dependency functions often prevent them from producing correct implementations.

---

> > > ### Author Response · Authors · 2025-11-28
> > > **Response to Reviewer 3az5 (2/2)**
> > >
> > > Beyond the overall distribution, several repositories exhibit distinct error patterns that reveal more fine-grained model weaknesses. To illustrate this, we examine three representative repositories (pandas, sphinx, and transformers) and summarize their error distributions in the table below.
> > >
> > > |                  | pandas-dev/pandas | sphinx-doc/sphinx | huggingface/transformers |
> > > | ---------------- | ----------------- | ----------------- | ------------------------ |
> > > | % IndexError     | **81.43**         | 0.00              | 0.39                     |
> > > | % AssertionError | 5.00              | **78.76**         | 13.50                    |
> > > | % TypeError      | 0.71              | 0.44              | **27.83**                |
> > > | % NameError      | 0.36              | 3.54              | **20.37**                |
> > > | % AttributeError | 0.00              | 0.88              | 6.62                     |
> > > | % ImportError    | 3.93              | 0.44              | 3.73                     |
> > > | % ValueError     | 4.64              | 0.44              | 6.23                     |
> > > | % RuntimeError   | 0.00              | 0.00              | 7.71                     |
> > > | % Others         | 3.93              | 15.5              | 13.62                    |
> > >
> > > We found that different code repositories exhibit distinct distributions of error types. For example, Pandas primarily produces *IndexError*, Sphinx is dominated by *AssertionError*, and Transformers frequently encounters *TypeError* and *NameError*.Moreover, based on our manual inspection of agent execution traces, we observe the following issues:
> > >
> > > - **pandas**: The agent can generally match the provided interfaces, but many failures stem from incorrect handling of indexing semantics. For example, in the test case `test_nearest_upsample_with_limit`, the agent attempts to access `index - 2` in a DataFrame with only one row, reflecting a boundary-condition error during upsampling. Although the interface of function `_get_timestamp_range_edges` clearly describes the timestamp-alignment behavior, the agent fails to implement this logic correctly. **These errors indicate that the model struggles to implement the algorithms** **with complex reasoning**, contributing to the high rate of IndexError.
> > > - **Sphinx**: Most failures stem from incorrect formatting.  For example, in the test case `test_table_builder`, the agent generates a table border `'+------+------+'` instead of the expected `'+--------+--------+'`, reducing the column width from eight characters to six. Notably, the `Table` class docstring already provides a correct example with eight-character columns, indicating that the necessary information is readily available.  **This shows that the model’s ability to read and reason over** **repository** **documentation is limited**, explaining the high rate of AssertionError.
> > > - **Transformers:** Many failures arise from mismanaging API signatures, reflecting its library’s highly modular and deeply layered design. For example, when implementing `_batch_encode_plus()`, the agent incorrectly forwards the argument `is_pair` to a parent method that does not include it, despite the codebase explicitly exposes this mismatch between the child and parent classes. Similarly, in `get_vocab`, the agent returns a string instead of a dictionary, even though the base class clearly annotates the return type as `dict[str, int]` and documents `“Returns the vocabulary as a dictionary of token to index.”` **These mistakes highlight the model’s difficulty in understanding type annotations and method hierarchies**, contributing to frequent TypeError occurrences.
> > >
> > > Overall, these analyses reveal a clear trend: current models can often match the provided interfaces but still face substantial bottlenecks in implementing core functional logic, and these limitations manifest as different error types across repositories. Therefore, deeper fine-grained error analysis helps clarify capability boundaries and identify concrete directions for improving agents across different software systems.

---

### Official Review · Reviewer_oMnc · 2025-10-28

**Soundness:** 3
**Presentation:** 3
**Contribution:** 2
**Rating:** 4
**Confidence:** 3

**Summary:**

This paper proposes ACE Bench, a novel benchmark for evaluating coding agents. While existing benchmarks focus on evaluating PRs or bug fixs, ACE-Bench emphasizes assessing feature-level implementation capabilities of coding agents.

**Strengths:**

- Evaluating feature-level implementation is both novel and important. As evidenced by recent SWE-bench leaderboard results, modern coding agents can perform bug-fixing tasks with high accuracy. However, their capability to handle feature-level implementations remains largely unexplored. This paper addresses this gap by providing a benchmark specifically designed to evaluate this capability.

- The benchmark is designed with usability in mind. Given that evaluating the full set requires approximately one million tokens (with associated computational costs), the authors provide a "lite set" that reduces evaluation costs. Additionally, the "Passed Rate" metric (the average fraction of fail-to-pass tests passed per task) enables partial assessment of feature-level implementation capability.

**Weaknesses:**

- Allowing unrestricted library usage may enable agents to complete tasks by simply calling existing library functions, essentially testing library knowledge rather than implementation capability (The benchmark allows agents to use pip install to add arbitrary libraries (Figure 13)). While the authors prevent accessing ground-truth implementations via anti-cheating mechanisms,the policy on legitimate library usage remains unclear. The authors should clarify whether the evaluation assesses (a) the ability to select and leverage appropriate libraries as part of software development skills, or (b) pure implementation capability with a fixed set of libraries. This distinction is critical for interpreting what capabilities are actually being measured.

- The benchmark tasks are derived from commits created between May 2022 and September 2025, which overlaps substantially with the training periods of models (i.e. knowledge cutoff). Fig. 5 shows that task performance currently exhibits minimal dependence on commit time. However, as the authors acknowledge, the risk of data leakage may become more pronounced in the future. Therefore, continuous updates to the benchmark will be critical to maintain its validity as an evaluation tool.

- The benchmark only supports Python, limiting its generalizability to other programming languages.

**Questions:**

- I found it somewhat confusing that there exists another agent evaluation benchmark with the same name ACE Bench [Chen+ 2025]

[Chen+ 2025] "ACEBench: Who Wins the Match Point in Tool Usage?"

---

> ### Author Response · Authors · 2025-11-21
> **Response to Reviewer oMnc (1/2)**
>
> Many thanks to your valuable comments and questions, which help us a lot to improve our work. Our responses are given below.
>
> ---
>
> **Q1:** **The authors should clarify whether the evaluation assesses (a) the ability to select and** **leverage** **appropriate libraries as part of software development skills, or (b) pure implementation capability with a fixed set of libraries.**
>
> **A1:** This is a very insightful question. Briefly, our ACE-Bench allows agents to install and use external libraries, aiming to measure agents' capability to develop an entire complex feature in general from a user-centric perspective. This is a more practical and valuable view for real-world development. Accordingly, we use pass rate and resolved rate as evaluation metrics, which reflect how well user requirements are satisfied.
>
> Real-world development cannot be cleanly separated into “leveraging libraries” and “pure implementation,” as these aspects are typically intertwined. The choice between them depends on task requirements, constraints, and the level of abstraction. For instance, when implementing an activation function, one may use NumPy APIs for basic computations and compose them into a complex function. Such behavior could be interpreted as either as "leverage appropriate libraries" from the perspective of using numpy APIs or as "pure implementation" due to not directly call a existing API that implements the full activation function.
>
> **Our evaluation is thus outcome-oriented:** we focus on whether the agent can correctly complete the feature, regardless of its use of libraries or custom code. Notably, after examining the output code from different agents, we find they generally produce complete implementations rather than simply calling a pre-existing API that matches the task requirement.  More fine-grained metrics such as token usage, trajectory length, and error patterns would provide deeper insights and are left for future work, as current agents still face challenges in completing tasks reliably.
>
> ---
>
> **Q2:** **Does the commit range introduce potential data leakage due to model training overlap?**
>
> **A2:** Thank you for raising this concern. Although LLM may have been trained on the repositories we chose , our newly generated task formulation is something LLM has never seen before. When SWE-Bench was first released, its source repositories already existed, yet the benchmark remained highly challenging because the task formulation was new. Similarly, the low resolved rate of 7.5% of advanced LLM on our benchmark also suggests that our tasks remain challenging and are not simply memorized content.
>
> We also acknowledge the concern regarding potential future data leakage. Once publicly released, such leakage is difficult to avoid as newer models begin to absorb trajectories similar to those produced by ACE-Bench. To mitigate this, we plan to update the benchmark every six months with newly collected evaluation data to ensure its continued validity and difficulty.
>
> ---
>
> **Q3: The benchmark only supports Python, limiting its generalizability to other programming languages.**
>
> **A3:** Thank you for this insightful comment. In the current version, we focus on Python because of its extensive adoption in the open-source ecosystem and the abundance of high-quality repositories with comprehensive test suites. Many existing repository-level benchmarks, such as SWE-Bench and its variants, are also built on Python, which facilitates comparison and ensures consistency with prior work. Moreover, Python’s familiarity among AI researchers makes manual validation and quality control more reliable.
>
> Previous studies have shown that model performance on Python benchmarks strongly correlates with results in other programming languages [1,2], suggesting that Python remains a meaningful proxy for assessing general coding ability.
>
> Our pipeline is modular and can be readily adapted to other languages with minimal modification—for instance, replacing pytest with Junit and the Python tracing component with a lightweight Java agent. Supporting multiple languages is an important next step toward evaluating cross-language generalization and will be a key direction for future development of ACE-Bench.
>
> [1] Rozière, Baptiste, et al. "Code Llama: Open Foundation Models for Code.", 2023, http://arxiv.org/abs/2308.12950. arXiv.
>
> [2] Cassano, Federico, et al. "MultiPL-E: A Scalable and Extensible Approach to Benchmarking Neural Code Generation", 2022, http://arxiv.org/abs/2208.08227. arXiv.

---

> ### Author Response · Authors · 2025-11-21
> **Response to Reviewer oMnc (2/2)**
>
> **Q4:** **I found it somewhat confusing that there exists another agent evaluation benchmark with the same name ACEBench [Chen+ 2025]**
>
> **A4:** Thank you for pointing this out, and we apologize for the naming oversight. Our benchmark is entirely independent of ACEBench proposed by Chen et al. (2025), and the two works address fundamentally different research problems.
>
> - **Different goal.** Our ACE-Bench evaluates agents on end-to-end feature development in real codebases, whereas Chen et al. study LLM tool-use behavior.
> - **Different methodology.** We rely on execution based evaluation in full repositories, while Chen et al. evaluate tool call sequences using AST and process level metrics.
> - **Different data generation.** Our tasks are generated automatically from real repositories, whereas theirs are constructed through human designed templates and scripted variations.
>
> In summary, the two benchmarks merely share an acronym but belong to different subfields. In accordance with the conference policy, we are fully open to renaming our benchmark if necessary, and we will coordinate with the area chair during the revision process to finalize a suitable alternative once all parties have agreed.

---

> ### Comment · Reviewer_oMnc · 2025-11-25
>
> Thank you for the response!
> I now clearly understand that implementation capability and the ability to effectively use libraries cannot be simply separated, and that the evaluation is fully outcome-oriented. While this evaluation may favor agents with broader library knowledge over those with pure implementation skills, this is a natural consequence of pursuing practical evaluation.That said, I still remain concerned about Q3. Currently, only Python is supported. So I will raise my score in anticipation of future extensions (I think this is particularly important for outcome-oriented evaluation).

---

> > ### Author Response · Authors · 2025-11-25
> > **Thanks so much for your feedback and raising the score!**
> >
> > Thank you so much for your positive feedback and constructive comments! We are delighted that our clarifications have effectively addressed your concerns regarding the evaluation objective. As you noted, ACE-Bench is inherently outcome-oriented to reflect real-world application scenarios.
> >
> > Regarding Q3, we fully agree that supporting multi-language is both reasonable and valuable. The extension of multi-language support has been included in our future development plans. We sincerely appreciate your understanding and your willingness to adjust your score in anticipation of future extensions.

---

### Official Review · Reviewer_erVP · 2025-10-31

**Soundness:** 2
**Presentation:** 4
**Contribution:** 2
**Rating:** 4
**Confidence:** 5

**Summary:**

The authors propose ACE-Bench a python-only, execution-based coding benchmark to evaluate coding agents' performance on feature development. Similar to related benchmarks, they (manually) create execution environments and extract fail2pass and pass2pass tests to evaluate whether a solution would solve the problem at hand without breaking other functionality. Problem statements are synthesized using a LLM and include invocation path, function signature (including input and output variables), as well as annotations. The authors developed an algorithm to extract functions that are relevant to a test patch from an object dependency graph. In the evaluation of the OpenHands agent with four LLMs shows that compared with SWE-Bench the resolve rates are much lower.

**Strengths:**

* The authors don't base their dataset on already existing ones but scrape their own data which lessens the risk of data leakage
* The paper is well written and easy to follow. Visualization illustrate the core aspects of the work well.
* Assessing feature development capabilities is an important area which is under-explored
* The dataset is seems to be significantly more complex in terms of gold solution lines, files, functions and number of tests.
* The graph-based function extraction is novel and seems sound

**Weaknesses:**

* The authors do not provide a lot of analysis to show that their tasks are truly solvable. Given that the problem statements are LLM generated, this needs to be shown. The authors propose that AssertionErrors indicate problem statements contain sufficient information. However, runnable code does not correlate with solvability of the tasks.
* The data set is Python only which severely limits to which degree one can measure coding agent performance.
* Only a single agent (OpenHands) is evaluated hence the claim of "laziness" is specific to their scaffold.
* All "feature development" tasks are more feature extensions rather than new features. While this is due to the fact that truly new features are extremely hard to test, this is a major limitation for a benchmark that focuses on feature development assessment.
* The quality of the generated prompts is hard to quantify, yet the authors claim to have developed a "high-quality" data set.
* Providing invocation path, function signature (including input and output variables), as well as annotations in the prompt seems unreasonable and not typical of a feature development task. More typical are natural language description that are rather vague.
* You consider SWE-Bench a benchmark that doesn't contain any feature requests. This is not true. As quantified by Rashid et al.[1] (containing 22% feature requests), it actually contains 18% feature requests.

1. Rashid, M. S., Bock, C., Zhuang, Y., Buchholz, A., Esler, T., Valentin, S., ... & Callot, L. (2025). SWE-PolyBench: A multi-language benchmark for repository level evaluation of coding agents. arXiv preprint arXiv:2504.08703.

**Questions:**

* You report 889 executable environments but only 212 tasks, can you explain the descrepancy?
* You configure the nnumber P2P tests. Doesn't this mean a there may be tests that would fail even though it would pass both P2P and F2P?
* Is the subset of 30 instances purely random or stratified in some way?
* In section 4.2.1 you say you conducted a "professional-level algorithm engineer" who revised prompts. Can you detail what this title means and how they were revised?
* I don't quite understand how you arrived at the $L_1$ vs. $L_2$ datasets. Can you elaborate?

---

> ### Author Response · Authors · 2025-11-21
> **Response to Reviewer erVP (1/3)**
>
> We sincerely thank the reviewer for the detailed and constructive feedback. We address the listed weaknesses and questions point by point below.
>
> ---
>
> **Q1:** **Are ACE-Bench tasks truly solvable given LLM-generated problem statements?** **/ How is the “high quality” of the LLM-generated prompts justified and quantified?**
>
> **A1:** Thank you for raising this concern. The solvability of ACE-Bench tasks is not determined by LLM-generated components. The core content of each problem statement—such as function signatures, interface definitions, invocation paths, and code comments—comes directly from the original repository rather than from the LLM. As described in Section 3.2 (“Problem Statement Generation”), the LLM is only used to add brief docstring-style comments in cases where the original function descriptions are missing. Notably, the repository’s native detailed interface information greatly reduces the ambiguity of the problems and enhances their solvability.
>
> To evaluate the solvability of our benchmark, we performed a manual verification on the lite dataset. Conducting feature-level manual verification (often involving more than 1,000 lines of code) is highly labor-intensive. For the lite set, each case takes approximately 1.1 hour to verify, totaling about 33 person-hours by a senior algorithm engineer. The table below compares the non-verified variant with the human-refined version in terms of solvability. The nearly identical agent performance between the two versions suggests that our tasks are both reliable and solvable.
>
> | Setting        | Models          | % Resolved | % Passed |
> | -------------- | --------------- | ---------- | -------- |
> | Original       | Gemini 2.5 Pro  | 3.3        | 17.0     |
> |                | GPT 5           | 6.7        | 29.5     |
> |                | Claude 4 Sonnet | 6.7        | 37.0     |
> | Human-Verified | Gemini 2.5 Pro  | 3.3        | 15.4     |
> |                | GPT 5           | 6.7        | 27.4     |
> |                | Claude 4 Sonnet | 3.3        | 32.5     |
>
> To ensure full reliability, we promise all the tasks in the released version will be verified.
>
> ---
>
> **Q2:** **The data set is Python only which severely limits to which degree one can measure coding agent performance.**
>
> **A2:** Thank you for this insightful comment.  For the current version, we focus on Python due to its wide usage and representativeness.
>
> - The current release focuses on Python due to its widespread use in the open-source ecosystem and the availability of high-quality repositories with comprehensive test suites. Many existing repository level benchmarks, including SWE Bench and its variants, also use Python. This choice makes the first version of ACE-Bench easier to adopt and compare with prior work. Additionally, Python is also one of the languages most familiar to AI practitioners, which makes manual inspection and quality control more reliable.
> - Prior work shows that model performance on Python benchmarks strongly correlates with performance in other languages [1,2], indicating that Python evaluation remains a useful proxy for general coding ability.
>
> Our pipeline is modular, allowing core components to be easily adapted to other languages with minimal changes. For example, extending to Java would involve replacing pytest-based test discovery with JUnit and using a small Java agent during mvn test, instead of Python tracing. Multi-language evaluation is essential for studying cross-language generalization and will be a valuable direction for future extensions of ACE-Bench.
>
> [1] Rozière, Baptiste, et al. "Code Llama: Open Foundation Models for Code.", 2023, http://arxiv.org/abs/2308.12950. arXiv.
>
> [2] Cassano, Federico, et al. "MultiPL-E: A Scalable and Extensible Approach to Benchmarking Neural Code Generation", 2022, http://arxiv.org/abs/2208.08227. arXiv.
>
> ---
>
> **Q3: Only a single agent (OpenHands) is evaluated hence the claim of "laziness" is specific to their scaffold.**
>
> **A3:** Thank you for the helpful comment. To investigate whether the observed "laziness" behavior is due to the scaffold, we conducted additional experiments on the lite subset using both Claude Code and Mini-SWE-Agent.  The table below reports the results. We observed the same tendency in both scaffolds, such as guessing interfaces instead of reading source files, indicating that this behavior is not tied to OpenHands.
>
> | Scaffold                         | % Resolved | % Passed |
>   | -------------------------------- | ---------- | -------- |
>   | Claude Code + Claude 4 Sonnet    | 6.7        | 40.5     |
>   | OpenHands + Claude 4 Sonnet      | 6.7        | 37.0     |
>   | Mini-SWE-Agent + Claude 4 Sonnet | 0.0        | 28.2     |

---

> > ### Comment · Reviewer_erVP · 2025-11-23
> > **Thanks**
> >
> > Thank you for your detailed response. As a follow up: For  how many instances did you have to generate the docstring using an LLM and how did you measure the reliability of the LLM-based approach in the second patch extraction step?

---

> ### Author Response · Authors · 2025-11-21
> **Response to Reviewer erVP (2/3)**
>
> **Q4:** **Are ACE-Bench tasks “feature extensions” rather than genuinely new features?**
>
> **A4:** ACE-Bench can generate genuinely novel feature tasks by deriving them from recent commits that postdate the training period of current language models. For an LLM trained before the specified start time, any functionality introduced afterward constitutes a new feature.
>
> -  For example, commit  1709ed9 in the *Transformers* repository (November 12, 2025) introduced the AudioFlamingo3 feature. For any LLM trained before that date, AudioFlamingo3 represents unseen functionality. By constructing tasks from such recent commits, ACE-Bench evaluates an agent’s ability to implement truly new features rather than extensions of existing ones.
>
> ---
>
> **Q5:** **Why does the prompt include invocation paths, signatures, and annotations instead of using vague natural-language descriptions typical of feature development?**
>
> **A5:** Thank you for raising this question. We adopt a structured specification to ensure both practical relevance and reliable evaluation, as detailed below.
>
> - Structured interface information reflects how feature development is organized in many real engineering workflows. This interface-driven setup naturally aligns with practical software development practices.
> - Clearly defined interfaces are also crucial for evaluation. High-level natural language descriptions often introduce ambiguity, making reliable benchmarking difficult. By using explicit signatures and invocation paths, we reduce uncertainty and improve task solvability, which makes this setup more suitable for the current stage of AI coding development.
>
> Developing features directly from vague natural language remains an important research direction. However, bridging the gap between abstract intent, concrete interfaces, and executable implementations remains beyond current LLM-based agents. ACE-Bench therefore targets this critical intermediate step, providing a rigorous and practically grounded foundation for future progress.
>
> ---
>
> **Q6:** **Isn’t it inaccurate to claim SWE-Bench has no feature requests, given Rashid et al. report 18–22% feature requests?**
>
> **A6:** We appreciate the reviewer’s comment and apologize for the confusion. While PR-based methods can capture certain feature-level tasks, their effectiveness is fundamentally limited by the quality and completeness of human-written PRs. In real-world repositories, a single feature is typically implemented across multiple PRs, followed by additional patches or bug fixes. Consequently, PR-based approaches often yield incomplete or noisy feature implementations, covering only a small subset of all features in the repository. In contrast, our method overcomes these inherent limitations, allowing systematic and large-scale extraction of complete, high-quality feature-level tasks.
>
> ---
>
> **Q7:** **You report 889 executable environments but only 212 tasks, can you explain the descrepancy?**
>
> **A7:** The pipeline generates 889 candidate tasks from 16 repositories. Applying benchmark filters on code size, fail-to-pass test points, and test activity yields 212 final tasks.
>
> ---
>
> **Q8:** **Configuring the number of** **P2P** **tests means there may be tests that fail even if both P2P and F2P pass.**
>
> **A8:**  This limitation is shared by our work and the SWE-bench series, stemming from the trade-off between efficiency and the breadth of P2P test coverage. Running full-repository tests for every instance would make both data generation and evaluation prohibitively expensive. For example, executing all unit tests in the *transformers* library takes about one day on two A100 GPUs. Therefore, it is impractical to impose no limit on the number of P2Ps. As a reference, SWE-bench reports an average of 111.7 P2P test points. Under our setting with five P2P files, the average number of P2P test points is 314.0, offering broader coverage while keeping the pipeline computationally feasible.

---

> > ### Comment · Reviewer_erVP · 2025-11-25
> > **Evaluation**
> >
> > In your example of the transformers commit 1709ed9, how do you make sure that an LLM implementation which is functionally correct is considered correct by your harness if it uses different function and variable names than the ones that the tests expect? My understanding is that all tests would fail because they can't find the correct functions to test.

---

> > > ### Author Response · Authors · 2025-11-26
> > > **Response to Reviewer erVP (1/2)**
> > >
> > > Thank you for your thoughtful feedback.  We have already considered this issue and accordingly incorporated detailed function interfaces into the problem statement. For commit 1709ed9, an agent can pass the tests as long as it implements the logic correctly under the provided interfaces. Below, we explain why providing interfaces is sufficient for evaluating feature-level software development and present additional experimental results showing that our ACE-Bench serves as a meaningful benchmark for assessing current code agents.
> > >
> > > - **Our interface already provides sufficient information for passing the unit tests.** Specifically, it includes class definitions, docstrings, class attributes, and method signatures, covering the “function and variable names” on which the tests depend. Taking commit 1709ed9 as an example, the interface for `AudioFlamingo3ForConditionalGeneration` is provided, including `init`, `forward`, and other required methods. Furthermore, its docstring explicitly includes the usage of `from_pretrained("nvidia/audio-flamingo-3")`, which effectively guides model development by clarifying potentially ambiguous implementation details such as layer and function naming conventions. Thus, as long as the code agent implements new features in accordance with the provided interface, it will successfully pass the corresponding unit tests. *Conversely, failure to do so despite such detailed interfaces indicates a lack of compliance, which constitutes an essential dimension in assessing a model’s capability.*

---

> > > ### Author Response · Authors · 2025-11-26
> > > **Response to Reviewer erVP (2/2)**
> > >
> > > - **The experiments demonstrate that our benchmark effectively evaluates the capability boundaries of cutting-edge code agents, offering actionable** **insights** **for future development.** By offering standardized interfaces, we ensure that functionally correct feature implementations are not penalized due to signature mismatches, an aspect overlooked by SWE-Bench. To quantify this effect, we perform an ablation study on the lite subset, comparing performance with and without the provided interfaces.
> > >
> > >   |                 | Without Interface |          | With Interface |          |
> > >   | --------------- | ----------------- | -------- | -------------- | -------- |
> > >   | Model           | % Resolved        | % Passed | % Resolved     | % Passed |
> > >   | Gemini 2.5 Pro  | 0.0               | 12.0     | 3.3            | 17.0     |
> > >   | OpenAI o3       | 3.3               | 17.7     | 3.3            | 25.6     |
> > >   | GPT-5           | 3.3               | 16.6     | 6.7            | 29.5     |
> > >   | Claude Sonnet 4 | 3.3               | 16.9     | 6.7            | 37.0     |
> > >
> > >   The result of pass rate indicates that, in the absence of interfaces, different models converge to similarly low performance levels, exhibiting minimal distinguishability, contradicting their observed differences in practical use. To further validate this conclusion, we evaluate a broader set of models on the lite subset with interfaces, including the recently released Gemini 3 Pro and Claude Opus 4.5 models from just a few days ago.
> > >
> > >
> > >   | Agent Scaffold | Model                          | Release Date | %Passed | %Resolved |
> > >   | -------------- | ------------------------------ | ------------ | ------- | --------- |
> > >   | Claude Code    | **Claude** **Opus** **4.5**    | 2025-11-25   | 60.3    | 20.0      |
> > >   |                | Claude Sonnet 4.5              | 2025-09-30   | 47.1    | 16.7      |
> > >   |                | Claude Sonnet 4                | 2025-05-23   | 40.5    | 6.7       |
> > >   | Gemini CLI     | **Gemini 3** **Pro**           | 2025-11-18   | 44.4    | 10.0      |
> > >   | OpenHands      | Claude Sonnet 4.5              | 2025-09-30   | 37.0    | 10.0      |
> > >   |                | Claude Sonnet 4                | 2025-05-23   | 37.0    | 6.7       |
> > >   |                | GPT-5                          | 2025-08-07   | 29.5    | 6.7       |
> > >   |                | Qwen3-Coder-480B-A35B-Instruct | 2025-07-23   | 25.6    | 0.0       |
> > >   |                | OpenAI o3                      | 2025-04-16   | 23.2    | 3.3       |
> > >   |                | Gemini 2.5 Pro                 | 2025-06-05   | 17.0    | 3.3       |
> > >   | mini-SWE-agent | Claude Sonnet 4.5              | 2025-09-30   | 28.2    | 0.0       |
> > >
> > >   With interfaces, performance differences become clear: stronger models consistently achieve higher pass and resolution rates (e.g., Claude Code + Claude Sonnet 4 / Sonnet 4.5 / Opus 4.5), aligning with the development timeline of code agents and confirming that the benchmark captures genuine capability distinctions.  Based on our benchmark results, we further identify several noteworthy phenomena:
> > >
> > >   - **Scaffold sensitivity.** The choice of scaffold can significantly influence model performance, as evidenced by the comparison among Claude Code, OpenHands, and mini-SWE-agent under Claude Sonnet 4.5.
> > >   - **Claude is deeply integrated with Claude Code.** After the release of Claude Code, the agentic coding capabilities of the Claude model series have become increasingly intertwined with this framework. Claude Sonnet 4 performs similarly on OpenHands and Claude Code, whereas Claude 4.5 performs substantially better on Claude Code. Combined with the release timeline (Sonnet 4 < Claude Code < Claude 4.5), this may suggest that Anthropic will henceforth focus on enhancing the agentic coding capabilities of the Claude models within the Claude Code framework, rather than optimizing their generality across other agent frameworks.
> > >
> > > Overall, these findings confirm that providing interfaces effectively eliminates ambiguity and enables meaningful, discriminative evaluation. This design choice ensures that ACE-Bench faithfully exposes the capability boundaries of current state-of-the-art agents and offers clear guidance for future progress in code-agent development.

---

> > > > ### Comment · Reviewer_erVP · 2025-11-26
> > > > **Increasing score**
> > > >
> > > > I thank the authors for addressing all my concerns. I especially like that novel feature development can be tested by providing interfaces. I will raise my score.

---

> > > > > ### Author Response · Authors · 2025-11-27
> > > > > **Thanks so much for your feedback and raising the score!**
> > > > >
> > > > > We sincerely thank the reviewer for the encouraging feedback and for raising the score. We are glad that our clarifications on interface-based feature development have effectively addressed the earlier concerns, and we deeply appreciate the acknowledgment of ACE-Bench’s novelty and its value in advancing the evaluation of code-agent capabilities.

---

> ### Author Response · Authors · 2025-11-21
> **Response to Reviewer erVP (3/3)**
>
> **Q9:** Is the subset of 30 instances purely random or stratified in some way?
>
> **A9:** We first sampled one instance from each repository to ensure diversity across codebases. The remaining instances were then randomly sampled from the full set.
>
> ---
>
> **Q10:** Can you detail what "professional-level algorithm engineer" title means and how they were revised?
>
> **A10:** We invited a senior engineer with five years of industry experience in AI infrastructure and system architecture to manually verify the lite subset. The verification required 33 hours in total, with an average of 1.1 hours per instance. Following a fixed procedure, the engineer ensured that each task was solvable under the given specification. When missing or ambiguous information was encountered, minimal clarifications were added to guarantee that the interfaces were sufficiently specified for a valid solution. Additional discussions are included in Section 4.2.1 of the revised manuscript.
>
> ---
>
> **Q11: I don't quite understand how you arrived at the L1 vs. L2 datasets. Can you elaborate?**
>
> **A11:** We apologize for the ambiguity. L1 denotes incremental development within an existing codebase, while L2 refers to development from scratch.
>
> - L1: Starting from the F2P test file, we trace its execution using `pytest` and mask all invoked objects (functions, classes, methods) in the original codebase. The agent receives this masked repository and the task interface, and must complete the missing implementations to pass all tests.
> - L2: We modify the F2P test file so that imports of top-level objects are redirected to a standalone `agent_code` module (e.g., replacing `from liger_kernel.transformers.dyt import LigerDyT` with `from agent_code import LigerDyT`). The agent has no access to the original codebase; it is only given the interface and task description, and must reimplement the required functionality entirely from scratch.

---

> ### Author Response · Authors · 2025-11-24
> **Response to erVP**
>
> Thank you for your insightful questions. Regarding the docstrings generated by the LLM, they account for 20% of the total. Regarding the reliability of the LLM-based classification of top-level objects, our pipeline’s post-verification mechanism ensures that cases are handled correctly.
>
>   - **In particular, the LLM is used solely to automate the second patch extraction step, while the subsequent post-verification mechanism effectively identifies and filters out incorrect classifications.** For instance, if the LLM selects objects unrelated to the intended feature, removing these objects would disrupt other functionality, causing the task to fail the P2P verification and thus be discarded. Conversely, if the LLM overlooks some feature-related objects, insufficient code would be removed, resulting in an F2P verification accuracy exceeding the threshold; such tasks also fail the F2P verification and are discarded.
>
> To ensure complete reliability, we guarantee that all tasks included in our benchmark undergo expert verification.

---

> > ### Comment · Reviewer_erVP · 2025-11-26
> > **Performance on 20% LLM-generated doc**
> >
> > It'd be interesting to see if whether there is a performance discrepancy between the 20% of tasks that have LLM-generated docs vs. the rest with existing doc.

---

> ### Author Response · Authors · 2025-11-27
> **Response to Reviewer erVP**
>
> Thank you for this valuable suggestion. Quantitatively analyzing the impact of using LLM-generated docstrings is indeed reasonable. However, directly comparing two non-overlapping sets of tasks (i.e., instances with and without such docstrings) has inherent limitations, as differences in task difficulty may confound the results. A more appropriate approach is to perform ablation studies solely on the instance set that includes these docstrings, evaluating performance under two conditions: with and without the LLM-generated docstrings. The results are shown below.
>
> |                                 | Removing LLM-generated docstrings      |  $\hspace{3em}$ Retaining LLM-generated docstrings    |
> | ------------------------------- | -------------------------------------- | ------------------------------------- |
> | Scaffold + Model                | %Passed $\hspace{6em}$ %Resolved   |  $\hspace{3em}$ %Passed $\hspace{6em}$ %Resolved  |
> | Claude Code + Claude Opus 4.5   | 21.9 $\hspace{8em}$ 18.2  |  $\hspace{3em}$ 57.0 $\hspace{8em}$ 27.3 |
> | Claude Code + Claude Sonnet 4.5 | 7.7 $\hspace{8.7em}$ 0.0 |  $\hspace{3em}$ 42.4 $\hspace{8em}$ 18.2 |
> | Claude Code + Claude Sonnet 4   | 15.6 $\hspace{8.2em}$ 9.1  |  $\hspace{3em}$ 36.4 $\hspace{8.2em}$ 9.1 |
>
> **The results reveal two clear findings: overall performance drops noticeably when LLM-generated docstrings are removed, and the ability to distinguish differences between models is substantially reduced.** By inspecting the error logs, we found that most failures stem from missing or incomplete information.
>
> For instance, in a task involving `PrettyPrinter().pformat`, the agent produced the Python string `'deque([1, 2])'` instead of the correctly formatted `'deque([\n\t1,\n\t2,\n])'` when given the Python object `deque([1, 2])`. This occurred because the native interface lacks a docstring, causing the model to overlook the indentation `\t` and line-break `\n` behavior. In contrast, the LLM-generated docstring explicitly states:
>
> > “This method converts any Python object into a formatted string with **proper indentation** and **line breaks** according to the PrettyPrinter's configuration settings (width, depth, and indent)...”
>
> The LLM generated this docstring correctly because the test function for this instance provides an input–output pair, where the input is the Python object `deque([1, 2])` and the expected output is the string `'deque([\n\t1,\n\t2,\n])'`. This docstring explicitly indicates that the agent must consider indentation and line breaks—details necessary to implement `PrettyPrinter().pformat`correctly. Therefore, supplementing the missing functional details is critical for ensuring accurate and meaningful evaluation.
>
> Furthermore, as noted in our response to Q1, we assessed the reliability of LLM-generated docstrings by asking a senior engineer to manually revise the prompts in the Lite set and then comparing the resulting performance. The closely matching results indicate that LLM-generated docstrings are sufficiently reliable for evaluation. Despite using LLMs to supplement missing docstrings for cost and scaling reasons, we assure that all released datasets will undergo manual verification to ensure their full reliability.

---

### Official Review · Reviewer_anv6 · 2025-11-01

**Soundness:** 3
**Presentation:** 3
**Contribution:** 2
**Rating:** 4
**Confidence:** 4

**Summary:**

This paper introduces ACE-Bench, an execution-based and continually updatable benchmark for feature-oriented agentic coding, built via a test-driven, dependency-trace pipeline that yields 212 tasks across 16 repositories and shows frontier agents solve only ~7.5%.

**Strengths:**

- The benchmark targets feature-level development (not just bug fixes) and pairs each task in two modes—L1 (extend an existing repo) and L2 (from scratch)—a clean formulation that isolates the role of context and raises the ceiling on task complexity.

- The evaluation is execution-based, with explicit interfaces and anti-cheating controls; the pipeline includes post-verification and ablations (e.g., hiding interfaces, step budgets, visible tests), plus clear metrics (Resolved/Passed/Token I/O). These choices make the results reproducible and the failure analysis informative.

- The paper is well structured (pipeline figures, instance layout, and evaluation workflow are easy to follow), and it surfaces useful empirical trends (e.g., performance drops with longer required code; L2 is markedly harder than L1).

**Weaknesses:**

- Positioning vs. closely related work needs to be sharper. The paper should more directly compare and differentiate from SWE-Dev (feature-driven development on large existing codebases with runnable environments; 14k train / 500 test and developer-authored unit tests) and commit0 (from-scratch library generation with API spec + interactive tests).

- Dataset composition skew. Although spanning 16 repos, the task mass is concentrated (e.g., Transformers dominates), which risks domain bias and may understate generalization across diverse stacks (services, infra, build systems).

- Baseline coverage & fairness details. All agents are run inside OpenHands, which is reasonable, but diversity in agent frameworks (gemini-cli, kimi-cli) would triangulate where the difficulty lies.

```
SWE-Dev: https://arxiv.org/abs/2505.16975
commit0: https://arxiv.org/abs/2412.01769
```

**Questions:**

Precise distinction from SWE-Dev and commit0.

---

> ### Author Response · Authors · 2025-11-21
> **Response to Reviewer anv6 (1/2)**
>
> We sincerely thank the reviewer for providing thoughtful reviews and constructive suggestions, which help us a lot to improve our work. We answer the questions as follows.
>
> ---
>
> **Q1: The paper should more directly compare and differentiate from SWE-Dev.**
>
> **A1:** We thank the reviewer for pointing out this interesting paper. We have updated the manuscript Appendix D to explicitly compare ACE-Bench with SWE-Dev. The main distinctions are summarized below.
>
> | Benchmark | Task Source | F2P/P2P | Real-world software development | Agent Evaluation Included | Avg. LoC |
>   | --------- | ----------- | ------- | ------------------------------- | ------------------------- | -------- |
>   | SWE-Bench | PR          | ✓       | ✓                               | ✗                         | 32.8     |
>   | SWE-Dev   | Unit Tests  | ✗       | ✗                               | ✗                         | 190      |
>   | ACE-Bench | Unit Tests  | ✓       | ✓                               | ✓                         | **1012** |
>
> - **Closer to real-world development and stricter data construction.** Each task in our benchmark preserves the originally well-developed features of the codebase while leaving the target features unimplemented, thereby more faithfully reflecting real-world incremental development workflows. This is achieved through a rigorous construction pipeline involving precise code patch extraction and strict post-verification (including both fail-to-pass and pass-to-pass checks). In contrast, SWE-Dev overlooks this aspect and omits pass-to-pass verification, which deviates from realistic development scenarios and may lead to solution patches that inadvertently break previously existing features.
>
> - **Interface-driven task specification to minimize ambiguity.** SWE-Dev uses LLM-generated project requirement description (PRD), which naturally introduces ambiguity. ACE-Bench instead exposes native top-level interfaces (i.e., function signatures and invocation paths) directly extracted from the original repositories, requiring agents to produce callable implementations. This design minimizes ambiguity, ensures implementability, and yields more stable and reliable evaluation outcomes.
>
> - **Agent-based experiments in this work.** While SWE-Dev evaluates a variety of LLM and multi-LLM setups, it does not report agent-based experiments. In ACE-Bench, we explicitly run experiments with end-to-end coding agents, using a unified OpenHands scaffold coupled with multiple LLM backends (e.g., Claude, GPT, Gemini, Qwen). This enables ACE-Bench to offer a more realistic view of how modern agents perform on feature-level development tasks and to serve as a stronger foundation for advancing future agent research.
>
> - **More realistic and complex feature-level tasks.** On average, SWE-Dev tasks require \~190 LoC across \~3 files. ACE-Bench tasks require \~1012 LoC across \~3.5 files and substantially more test points (\~447). This scale reflects the multi-file, cross-module modifications that typically arise in real feature development, rather than the more localized patches captured by SWE-Dev.
>
> ---
>
> **Q2: The paper should more directly compare and differentiate from commit0 (from-scratch library generation with** **API** **spec + interactive tests).**
>
> **A2:** Thank you for the suggestion. We have updated the manuscript Appendix D to explicitly contrast ACE-Bench with commit0. The main distinctions are summarized below.
>
> | Benchmark | Full Implementation | Realistic Software Development | Scalability |
>   | --------- | ------------------- | ------------------------------ | ----------- |
>   | commit0   | ✗                   | ✗                              | ✗           |
>   | ACE-Bench | ✓                   | ✓                              | ✓           |
>
> - **Real-world development task with full implementation.** In *commit0*, only the bodies of functions and classes are removed, while their definitions and the overall codebase architecture are retained. This effectively turns the task into a fill-in-the-blank exercise rather than a genuine from-scratch development problem. In contrast, ACE-Bench removes all implementations, granting the code agent substantially greater freedom and making the task much closer to real-world software development.
>
> - **Low-cost scaling to new repositories.** As noted in the limitations of *commit0's* main paper, it requires repositories with well-organized documentation and high unit-test coverage (>90%), which significantly limits the set of compatible libraries. In contrast, ACE-Bench only requires a runnable unit-test suite. After a quick three-minute configuration, the rest of the pipeline is fully automated. This design makes ACE-Bench far more scalable across a wider range of real-world codebases.

---

> ### Author Response · Authors · 2025-11-21
> **Response to Reviewer anv6 (2/2)**
>
> **Q3: Dataset composition skew.**
>
> **A3:** Thank you for this suggestion. Our initial repository selection mostly follows those used in SWE-Bench, whose repositories are relatively old. To reduce contamination risk and maintain realistic complexity, our benchmark requires test files committed after May 2022 and containing at least 100 lines of relevant features. Consequently, newer and more actively maintained projects, such as *Transformers*, naturally dominate our first-version dataset due to their diversed, high-quality implementations of cutting-edge algorithms spanning architecture, training, inference, and optimization, along with comprehensive test suites.
>
> **Such bias can be easily balanced.** We acknowledge that the current setup may reflect certain domain tendencies. However, that distribution can be easily balanced, as our pipeline scales efficiently in a fully automated manner. To demonstrate this, we selected 18 new repositories and, within two days, expanded the dataset with 638 additional cases covering services, infrastructure, and building systems. The expanded benchmark exhibits a more balanced and diverse composition, and the final public release will include a verified dataset with guaranteed diversity based on these new cases.
>
> | repositories                   | tasks | lines | files | objects | test points |
> | ------------------------------ | ----- | ----- | ----- | ------- | ----------- |
> | mlflow/mlflow                  | 280   | 600.3 | 13.6  | 25.8    | 11.7        |
> | optuna/optuna                  | 80    | 190.6 | 5.5   | 9.2     | 5.4         |
> | Lightning-AI/pytorch-lightning | 78    | 953.8 | 22.8  | 52.6    | 7.5         |
> | python/mypy                    | 30    | 457.2 | 7.7   | 20.2    | 16.3        |
> | SCons/scons                    | 26    | 137.7 | 3.9   | 7.6     | 7.9         |
> | fastapi/fastapi                | 20    | 409.3 | 5.3   | 6.6     | 4.0         |
> | pypa/hatch                     | 18    | 391.0 | 9.0   | 24.9    | 15.4        |
> | pypa/setuptools                | 18    | 869.5 | 13.9  | 55.4    | 9.9         |
> | pypa/virtualenv                | 18    | 66.6  | 2.9   | 7.6     | 5.6         |
> | pydantic/pydantic              | 12    | 495.1 | 7.1   | 23.4    | 17.3        |
> | huggingface/accelerate         | 12    | 423.4 | 7.5   | 10.8    | 13.4        |
> | Netflix/metaflow               | 10    | 171.8 | 2.8   | 4.6     | 8.2         |
> | hpcaitech/ColossalAI           | 8     | 141.4 | 3.1   | 3.0     | 1.1         |
> | pypa/packaging                 | 8     | 484.3 | 3.4   | 25.3    | 62.0        |
> | platformio/platformio-core     | 6     | 249.2 | 5.6   | 13.4    | 5.2         |
> | deepseek-ai/smallpond          | 4     | 216.0 | 4.0   | 9.3     | 3.7         |
> | mesonbuild/meson               | 4     | 343.8 | 14.8  | 21.0    | 25.0        |
> | psf/black                      | 2     | 55.0  | 2.0   | 2.0     | 16.0        |
>
> ---
>
> **Q4: Baseline coverage and fairness across different agent frameworks.**
>
> **A4:** To examine whether the difficulty of ACE-Bench arises from the tasks themselves rather than from a particular agent scaffold, we conducted additional experiments on the lite subset using both Claude Code and Mini-SWE-Agent. The results are shown below.
>
> | Scaffold                         | % Resolved | % Passed |
>   | -------------------------------- | ---------- | -------- |
>   | Claude Code + Claude 4 Sonnet    | 6.7        | 40.5     |
>   | OpenHands + Claude 4 Sonnet      | 6.7        | 37.0     |
>   | Mini-SWE-Agent + Claude 4 Sonnet | 0.0        | 28.2     |
>
> These additional experiments show trends consistent with those observed under OpenHands, even when using the strongest agent and LLM combinations, suggesting that the challenge primarily stems from the task design itself rather than from any specific agent framework. We have included these results and a short discussion in the revised manuscript (Section 4.2.2).

---

> ### Author Response · Authors · 2025-11-28
> **Looking forward to the reviewer’s valuable feedback. :)**
>
> Dear Reviewer anv6,
>
> We sincerely thank you again for your thoughtful and constructive feedback. In our latest revision, we have added new experiments and analyses based on your suggestions. We would greatly appreciate it if you could let us know whether these revisions address your concerns.
>
> Best regards,
> Authors

---

### Author Response · Authors · 2025-12-02
**Summary of Rebuttal Process and Score Improvements**

Dear Area Chair,

We sincerely appreciate your time and effort in handling our submission.

Notably, before the recent ICLR system issues occurred, we had already engaged in productive and detailed discussions with all reviewers and effectively addressed the major concerns they raised. These exchanges led to positive feedback, with multiple reviewers explicitly acknowledging that their concerns had been resolved.  As a result, **two reviewers confirmed that they had raised their scores**, bringing the overall score profile to  **4, 6, 6, 6** (from the initial 4, 4, 4, 6).

Specifically:

- **Reviewer anv6**: Detailed responses fully addressing the reviewer's all concerns have been presented, though due to the unexpected early end of the discussion phase, the reviewer is *unable to give further comments*. For your convenience, we highlight the main points as follows:
  - **Comparison with SWE-Dev and commit0**: The **SWE-Dev** lacks P2P unit test checks and interface definition, making evaluation unreliable and controversial. Moreover, it features only *plain multi-turn chat-based* evaluation, while our ACE-Bench evaluates *agents* (e.g. OpenHands). The **commit0** is more like *fill-in-the-blank* exercises and has many restrictions on repositories for properly building test data, while our work only requires a runnable test suite to build feature-level development tasks. Overall, our ACE-Bench is obviously more scientifically designed and built with better scalability.
  - **Dataset composition skew**: The reasons are clearly explained, since *Transformers* features numerous up-to-date, thoroughly tested, and well-maintained algorithm implementations that are suitable for feature-level software development. Moreover, 638 additional instances from 18 new repositories are provided to demonstrate that our ACE-Bench can be easily and quickly scaled and balanced.
  - **Task difficulty possibly caused by agent framework**: Results on two extra frameworks show consistent trends, demonstrating the difficulty of the tasks themselves.
- **Reviewer erVP (raised score):** We primarily addressed the reviewer’s concerns regarding task solvability and the reliability of interface-based evaluation. The reviewer acknowledged the value of our benchmark, noting:*“I thank the authors for addressing all my concerns. I especially like that novel feature development can be tested by providing interfaces. I will raise my score.”*
- **Reviewer oMnc (raised score):** Our clarifications successfully resolved the main concerns about the evaluation objective. The reviewer commented: *“I now clearly understand that the evaluation is fully outcome-oriented, so I will raise my score.”*
- **Reviewer ANuP (maintained score at 6):** The reviewer remained supportive and positive about the paper.

All reviewers who participated in the discussion recognized the significance of ACE-Bench in advancing the development of agentic coding. However, due to a recent system malfunction that reverted the rebuttal to earlier states, the final scores may not be correctly visible on the interface. We provide this clarification to ensure that the reviewers’ updated evaluation and our successful rebuttal receive appropriate consideration.

Thank you again for your consideration and understanding.

Best regards,

Authors

---

### Meta-Review · Area_Chair_bdqD · 2026-01-11

**Summary:**

The paper provides an agentic coding benchmark, and all reviewers recognized the significance of such as benchmark for advancing agentic coding. The paper should be selecting primary area as benchmark and datasets, rather than the current track, but it is minor and acceptable. Most major method-related questions were answered with supporting data. As a result, I am recommending to accept the paper, but would not mind if the decision gets bumped down.

**Reviewer Concerns:**

One concern is that positioning vs. SWE-Dev and commit0 not sharp enough, and Appendix D with explicit comparison table is very helpful. The authors did a good job on scale and diversity limitation brought up in the review, and add 600+ new cases in two days, with more detailed analysis on correlation analysis, interface matching, error-type classification, etc. These new analysis are suggested to be kept into the next version. Most reviewer concerns have been addressed through detailed responses, experiments, and clarifications. Two concerns remain as future work items are multi-language support and evaluation scenarios with vague natural-language requirements, which seems acceptable to me.

**Reviewer Scores:**

The scores before rebuttals are 4 4 4 6 and after rebuttals are 4 6 6 6. The authors' rebuttals show strong engagement with feedback, providing some new quantitative evidence and additional experiments to support their claims.

---

### Decision · Program_Chairs · 2026-01-26

Accept (Poster)